# META-WEIGHTED LANGUAGE MODEL TUNING FOR AUGMENTATION-ENHANCED FEW-SHOT LEARNING

## ABSTRACT

Recent studies have revealed the intriguing few-shot learning ability of pretrained language models (PLMs): They can quickly adapt to a new task when fine-tuned on a small amount of labeled data formulated as prompts, without requiring abundant task-specific annotations. Despite their promising performance, most existing few-shot approaches that only learn from the small training set still underperform fully supervised training by nontrivial margins. In this work, we study few-shot learning with PLMs from a different perspective: We first tune an autoregressive PLM on the few-shot samples and then use it as a generator to synthesize a large amount of novel training samples which augment the original training set. To encourage the generator to produce label-discriminative samples, we train it via weighted maximum likelihood where the weight of each token is automatically adjusted based on a discriminative meta-learning objective. A classification PLM can then be fine-tuned on both the few-shot and the synthetic samples with regularization for better generalization and stability. Our approach FewGen achieves an overall better result across seven classification tasks of the GLUE benchmark than existing few-shot learning methods, improving no-augmentation methods by 5+ average points, and outperforming augmentation methods by 3+ average points [1].

## 1 INTRODUCTION

Recent research has demonstrated the appealing few-shot learning potential of pretrained language models (PLMs) (Brown et al., 2020; Clark et al., 2020; Devlin et al., 2019; He et al., 2021; Liu et al., 2019; Meng et al., 2021) on natural language understanding (NLU) tasks (Wang et al., 2019; 2018): Instead of relying on abundant task-specific annotations, PLMs can effectively leverage a small set of training samples to quickly learn a new task. Such training data efficiency is usually achieved by formulating downstream tasks as prompts (Brown et al., 2020; Gao et al., 2021; Scao & Rush, 2021; Schick & Schütze, 2021a;d) which allow the PLM to adapt its language modeling ability acquired through pretraining to new downstream tasks.

The success of prompt-based methods has stimulated numerous explorations along the line of effective few-shot learning with PLMs: The training samples converted to natural language prompts can be used to directly fine-tune PLMs (Gao et al., 2021; Schick & Schütze, 2021a) or as in-context demonstrations to facilitate better inference (Brown et al., 2020; Liu et al., 2022b). More recent approaches aim to automate the design of prompts by gradient-based searching (Shin et al., 2020) or parameterizing prompts as continuous learnable embeddings (Lester et al., 2021; Liu et al., 2021b; Zhang et al., 2022; Zhong et al., 2021). Other studies investigate and address specific issues in prompt-based few-shot learning (Liu et al., 2022a; Tam et al., 2021; Zhao et al., 2021). While remarkable, the model performance still has a nontrivial gap from fully supervised models trained on massive labeled data. Indeed, training deep models is inherently data demanding—model generalization usually benefits from more training samples (Baum & Haussler, 1988).

In this work, we study few-shot learning with PLMs from a different perspective: Instead of proposing new methods for fine-tuning on few-shot samples, we focus on the generation of quality training data based on few-shot samples and using these synthesized training samples to fine-tune the classification models. Motivated by the strong text generation power of autoregressive PLMs (Brown et al., 2020;

---

[1]Code is shared in the supplementary material.

Keskar et al., 2019; Raffel et al., 2019), previous data augmentation methods enlarge the training set by synthesizing new samples based on the few-shot samples. They either fine-tune the generator on the training set with the standard maximum likelihood objective (Anaby-Tavor et al., 2020; Kumar et al., 2020) or use the training samples as demonstrations (Yoo et al., 2021). However, these methods do not explicitly model the distinction across different labels and may struggle to generate accurate training samples pertaining to the desired labels for challenging NLU tasks.

In this paper, we study how to use few-shot samples to effectively tune PLMs to generate high quality label-discriminative training samples. Our contributions are as follows: (1) We analyze the issues of using standard maximum likelihood for tuning the generator and propose a meta-weighted maximum likelihood objective for generator tuning by automatically learning token weights that emphasize label discriminativeness. (2) We propose a simple and effective training procedure for fine-tuning classification PLMs on generated data by mitigating label noise. (3) Under the same few-shot learning setting, our method FewGen outperforms existing methods by $3+$ average points on seven classification tasks of the GLUE benchmark (Wang et al., 2018). Ablation studies demonstrate the effectiveness of our proposed meta-weighted training objective and classifier fine-tuning method.

## 2 RELATED WORK

**Few-Shot Learning with PLMs.** Few-shot learning has gained much attention recently due to its minimal resource assumption—Without requiring massive annotated data but only leveraging a few training samples (*e.g.*, 16 per label), few-shot methods can be widely adopted in many practical scenarios where obtaining large-scale annotations is unaffordable. Standard fine-tuning of PLMs for few-shot learning usually performs poorly because the limited training samples may not be sufficient for optimizing the parameters in the newly introduced classification head. To reuse the language modeling ability of PLMs without introducing randomly initialized parameters, prompt-based approaches (Brown et al., 2020; Gao et al., 2021; Hu et al., 2022; Logan IV et al., 2021; Min et al., 2022; Schick & Schütze, 2021a;b;d; Tam et al., 2021) formulate training samples as natural language prompt templates so that various downsteam tasks can be solved as a token prediction problem. They enjoy improved training data efficiency over standard fine-tuning in low-data regimes (Scao & Rush, 2021) and achieve remarkable few-shot learning performance. Later developments in prompt-based methods replace the manual design of prompt templates with automatic search or learning (Cui et al., 2022; Hambardzumyan et al., 2021; Lester et al., 2021; Liu et al., 2021b; Zhang et al., 2022; Zhong et al., 2021). There are also studies focusing on specific issues in prompt-based methods such as densifying the supervision by revising the training objective (Liu et al., 2022a; Tam et al., 2021) and calibrating the biased predictions of PLMs before fine-tuning (Zhao et al., 2021). Instead of focusing on fine-tuning methods for few-shot learning, we study how to effectively generate abundant quality training samples by learning from the few-shot samples and use them to improve the generalization of the classification model.

**Data Augmentation.** Data augmentation methods (Chen et al., 2020; Lee et al., 2021; Miyato et al., 2017; Xie et al., 2020) aim to create similar samples to the existing ones so that the enlarged training set can benefit model generalization. Early approaches simply use manually designed rules (*e.g.*, swapping or inserting tokens) for word-level alterations over the given samples to create new ones (Wei & Zou, 2019). Later methods leverage the strong generation power of PLMs to synthesize novel samples from scratch. Given a training set, the PLMs can be either fine-tuned on the labeled samples to learn label-conditioned generation probability (Kumar et al., 2020; Lee et al., 2021; Yang et al., 2020) or take the labeled data as demonstrations (Wang et al., 2021; Yoo et al., 2021) to generate similar samples pertaining to the same label. In this work, we study how to effectively tune generators on few-shot training data for creating new data—standard fine-tuning of PLMs on a small set of training data is prone to overfitting, and the resulting model may struggle to generate accurate, diverse and novel training data. We address this challenge by leveraging prefix-tuning and proposing a new meta-weighted training objective to emphasize label-discriminative tokens for generator tuning.

**Controlled Text Generation.** Generating training samples for different labels can be viewed as a form of controlled text generation (Hu et al., 2017), whose goal is to generate textual contents of desired semantics, styles or attributes. Such control can be realized through different stages of PLM training and deployment: During pretraining, control codes (Keskar et al., 2019) can be used as

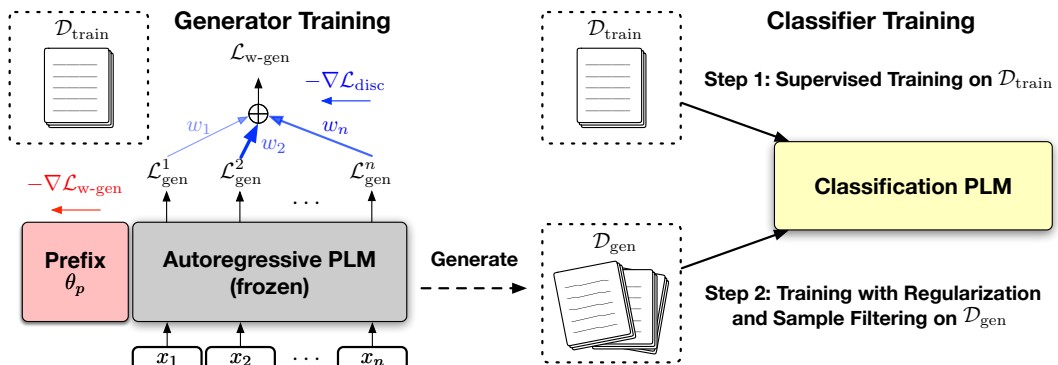

Figure 1: Overview of FewGen. A generator PLM is first tuned on the few-shot samples with our proposed meta-weighted maximum likelihood objective and then used to synthesize new training samples. A classification PLM is finally trained on both the few-shot and the generated samples.

explicit guidance for training the model to generate domain/attribute-specific texts; fine-tuning PLMs with attribute-specific data can also grant high-level control (*e.g.*, certain topics or sentiments (Ziegler et al., 2019)), fine-grained control (*e.g.*, specific words or phrases (Chan et al., 2021)) or both (Khalifa et al., 2021); at inference time, control over desired attributes can also be enforced without updating the PLM parameters (Dathathri et al., 2020; Krause et al., 2021; Kumar et al., 2021; Liu et al., 2021a; Pascual et al., 2021; Yang & Klein, 2021). Recently, a few studies explore fine-tuning autoregressive PLMs (Anaby-Tavor et al., 2020; Yang et al., 2020) with the standard language modeling objective on the training set or using label-specific prompts (Meng et al., 2022; Schick & Schütze, 2021c; Wang et al., 2021; Ye et al., 2022) to steer text generation towards the desired label.

**Meta-Learning for Sample Weighting.** The idea of weighting training samples in the loss calculation originates from the class imbalance (Wang et al., 2017) and noisy label (Hendrycks et al., 2018) learning scenarios—By assigning higher weights to the samples from minority classes or lower weights to the noisy samples, the learning process is less impacted by the imbalance/label noise issues. Meta-learning (Andrychowicz et al., 2016; Finn et al., 2017; Franceschi et al., 2018; Wu et al., 2018) is one way to automatically learn the weight for each sample. Specifically, a meta objective, usually defined as the loss on a clean unbiased validation set (Ren et al., 2018; Shu et al., 2019), can be used to learn the sample weights which become hyperparameters that control the optimization of model parameters. Our work has a different motivation and formulation of the meta objective for token-wise weighted training: Not all tokens in a training sample are equally label-discriminative. We thus design a meta objective to emphasize distinction across different labels (instead of using the validation loss as the meta objective) for learning the token weights.

## 3 METHOD

### 3.1 PRELIMINARIES

**Overview.** We consider the strict few-shot learning setting (Perez et al., 2021): The training set $\mathcal{D}_{\text{train}} = \{(\boldsymbol{x}, y)_i\}$ consists of $K$ training samples per label where $\boldsymbol{x} = [x_1, x_2, \ldots, x_n]$ is a text sequence with $n$ tokens. The development set $\mathcal{D}_{\text{dev}}$ is of the same size as $\mathcal{D}_{\text{train}}$. There is no access to additional task-specific unlabeled data. The number of training samples $K$ is assumed to be very small (*e.g.*, $K = 16$), making it challenging to train a classification model $C_{\phi}$ that generalizes well to unseen data. To mitigate such a training data scarcity issue, we propose to first train an autoregressive PLM on $\mathcal{D}_{\text{train}}$, and then use it as a generator $G_{\boldsymbol{\theta}}$ to synthesize a large amount of novel samples $\mathcal{D}_{\text{gen}} = \{(\tilde{\boldsymbol{x}}, \tilde{y})_i\}$ that augment the original training set. Finally, a classification PLM $C_{\phi}$ is fine-tuned on both $\mathcal{D}_{\text{train}}$ and $\mathcal{D}_{\text{gen}}$ to perform the task. An overview of our FewGen method is shown in Fig. 1.

**Text Generation with Autoregressive PLMs.** In standard fine-tuning for text generation, an autoregressive PLM $G_{\boldsymbol{\theta}}$ is trained via the maximum likelihood generation loss of each token in a

sequence $\boldsymbol{x}$ conditioned on previous tokens:

$$\min_{\boldsymbol{\theta}} -\frac{1}{n}\sum_{j=1}^{n}\log p_{\boldsymbol{\theta}}(x_j|\boldsymbol{x}_{<j}), \quad p_{\boldsymbol{\theta}}(x_j|\boldsymbol{x}_{<j}) = \frac{\exp(\boldsymbol{e}_j^{\top}\boldsymbol{h}_j)}{\sum_{j'=1}^{|V|}\exp(\boldsymbol{e}_{j'}^{\top}\boldsymbol{h}_j)}.$$

where the token generation probability $p_{\boldsymbol{\theta}}(\cdot)$ is usually parameterized using token embeddings $\boldsymbol{e}$ and hidden states $\boldsymbol{h}$ of a Transformer (Vaswani et al., 2017) model. After training, $G_{\boldsymbol{\theta}}$ can be used to generate novel texts by iteratively sampling tokens from its generation probability distribution.

**Prefix-Tuning.** Unlike fine-tuning which updates all model parameters $\boldsymbol{\theta}$ of a PLM, prefix-tuning (Li & Liang, 2021) freezes all pretrained Transformer parameters and only optimizes prefix vectors $\boldsymbol{\theta}_p$ that are prepended to each Transformer layer. We use prefix-tuning for training $G_{\boldsymbol{\theta}_p}$ on $\mathcal{D}_{\text{train}}$ because (1) it offers better effectiveness than fine-tuning for small datasets (Li & Liang, 2021) and (2) the generation models for different labels can share the same backbone Transformer parameters with only the prefix vectors being different, significantly reducing the memory requirement for multi-class classification tasks.

## 3.2 LABEL-DISCRIMINATIVE TEXT GENERATOR TUNING WITH META WEIGHTS

**Motivation.** To model the conditional text generation probability $p(\boldsymbol{x}|y_l)$ on different labels, a straightforward way is to parameterize a generation model $G_{\boldsymbol{\theta}_{p_l}}$ for each label $y_l$ via a set of prefix vectors $\boldsymbol{\theta}_p = \{\boldsymbol{\theta}_{p_l}\}|_{l=1}^{L}$ so that $p(\boldsymbol{x}|y_l) = p_{\boldsymbol{\theta}_{p_l}}(\boldsymbol{x})$, and then tune $\boldsymbol{\theta}_{p_l}$ on the training samples $\boldsymbol{x}$ with label $y_l$:

$$\min_{\boldsymbol{\theta}_{p_l}} \mathcal{L}_{\text{gen}}, \quad \mathcal{L}_{\text{gen}}(\boldsymbol{\theta}_{p_l}) = -\frac{1}{n}\sum_{j=1}^{n}\log p_{\boldsymbol{\theta}_{p_l}}(x_j|\boldsymbol{x}_{<j}). \tag{1}$$

However, such an approach only optimizes the *generative* likelihood $p(\boldsymbol{x}|y_l)$ without accounting for *label discriminativeness* $p(y_l|\boldsymbol{x})$ which is essential for generating unambiguous training samples to benefit the final classification task. Challenging NLU tasks can have largely similar distributions across different labels, with very nuanced differences reflected by a few key tokens. For example, a negative review text "*a movie where the ending feels like a cop-out*" may immediately become a positive one by just changing the last word "cop-out" to "revelation". Indeed, we find that such subtle distinctions over different labels may not be effectively captured by the generators if they are trained with the standard generation objective in Eq. (1). As shown in Fig. 2, $\mathcal{L}_{\text{disc}}$ (defined in Eq. (2)) can even increase during training—It is possible that the dominating patterns in the training samples are

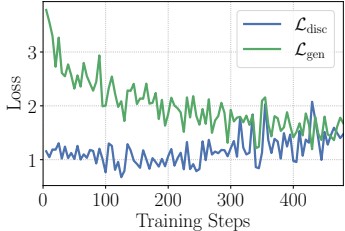

Figure 2: (On MNLI) Training the generator only via $\mathcal{L}_{\text{gen}}$ does not automatically decrease $\mathcal{L}_{\text{disc}}$.

label-indiscriminate (*e.g.*, a movie review dataset may frequently mention "the movie"), making the generators of different labels eventually converge to similar distributions, especially when there are limited training samples per label.

To promote the generation of label-discriminative texts, we encourage each token $x_j$ to be more likely generated under the corresponding label $y_l$ instead of other labels (*i.e.*, maximize $p_{\boldsymbol{\theta}_{p_l}}(x_j|\boldsymbol{x}_{<j})$ and minimize $p_{\boldsymbol{\theta}_{p_{l'}}}(x_j|\boldsymbol{x}_{<j})$ for $l' \neq l$) via a discriminative loss $\mathcal{L}_{\text{disc}}$:

$$\mathcal{L}_{\text{disc}}(\boldsymbol{\theta}_p) = -\frac{1}{n}\sum_{j=1}^{n}\mathcal{L}_{\text{disc}}^{j}(\boldsymbol{\theta}_p), \quad \mathcal{L}_{\text{disc}}^{j}(\boldsymbol{\theta}_p) = \frac{p_{\boldsymbol{\theta}_{p_l}}(x_j|\boldsymbol{x}_{<j})}{\sum_{l'=1}^{L}p_{\boldsymbol{\theta}_{p_{l'}}}(x_j|\boldsymbol{x}_{<j})} \tag{2}$$

Although one can directly combine $\mathcal{L}_{\text{disc}}$ with $\mathcal{L}_{\text{gen}}$ to train $G_{\boldsymbol{\theta}_p}$ to enforce distinction across different labels, doing so will result in two undesirable consequences: (1) A hyperparameter needs to be introduced to balance the weights of the two losses, whose optimal value is likely to vary by task; and (2) the generation-irrelevant loss $\mathcal{L}_{\text{disc}}$ will unavoidably interfere the language modeling process, making the resulting model prone to generating less fluent and coherent texts.

---

**Algorithm 1** Meta-Weighted Generator Tuning.

---

**Input:** $\mathcal{D}_{\text{train}}$: Few-shot training set.
**Parameter:** $T$: Number of training steps.
**Output:** $\boldsymbol{\theta}_p$: Prefix parameters for all labels.
Initialize $\boldsymbol{\theta}_p^{(0)}$ (with task-descriptive prompts) and $\boldsymbol{\omega}^{(0)}$
**for** $t \in [0, 1, \dots, T-1]$ **do**
> $\mathcal{B} \leftarrow$ Sample a minibatch from $\mathcal{D}_{\text{train}}$
> $\hat{\boldsymbol{\theta}}_p^{(t)}\left(\boldsymbol{\omega}^{(t)}\right) \leftarrow$ Take one gradient step to descend $\mathcal{L}_{\text{w-gen}}\left(\boldsymbol{\theta}_p^{(t)}; \boldsymbol{\omega}^{(t)}\right)$ on $\mathcal{B}$
> $\boldsymbol{\omega}^{(t+1)} \leftarrow$ Take one gradient step to descend $\mathcal{L}_{\text{disc}}\left(\hat{\boldsymbol{\theta}}_p^{(t)}\left(\boldsymbol{\omega}^{(t)}\right)\right)$ on $\mathcal{B}$
> $\boldsymbol{\theta}_p^{(t+1)} \leftarrow$ Take one gradient step to descend $\mathcal{L}_{\text{w-gen}}\left(\boldsymbol{\theta}_p^{(t)}; \boldsymbol{\omega}^{(t+1)}\right)$ on $\mathcal{B}$

**end**
**return** $\boldsymbol{\theta}_p = \boldsymbol{\theta}_p^{(T)}$

---

**Weighted Maximum Likelihood Generator Tuning.** To preserve the generative learning of $G_{\boldsymbol{\theta}_p}$ while emphasizing label-discriminative tokens, we assume each token is associated with a weight in the maximum likelihood loss. Intuitively, when our goal is to generate distinctive texts across different labels as training samples, not all tokens should contribute equally to generator training. For example, for sentiment classification tasks, one would expect "good/bad" to be more label-discriminative than "the movie", and the former should be paid more attention to during training. It is thus natural to generalize $\mathcal{L}_{\text{gen}}$ in Eq. (1) to $\mathcal{L}_{\text{w-gen}}$ as follows by assuming a weight $w_j$ is given for each token.

$$\min_{\boldsymbol{\theta}_{p_l}} \mathcal{L}_{\text{w-gen}}, \quad \mathcal{L}_{\text{w-gen}}(\boldsymbol{\theta}_{p_l}; \boldsymbol{w}) = -\sum_{j=1}^{n} w_j \mathcal{L}_{\text{gen}}^j(\boldsymbol{\theta}_{p_l}), \quad \mathcal{L}_{\text{gen}}^j(\boldsymbol{\theta}_{p_l}) = \log p_{\boldsymbol{\theta}_{p_l}}(x_j | \boldsymbol{x}_{<j}). \quad (3)$$

Note that in $\mathcal{L}_{\text{w-gen}}$, $\boldsymbol{w}$ is assumed to be the *hyperparameter* under which $\boldsymbol{\theta}_{p_l}$ is optimized. When $w_j$ is the same for every token, Eq. (3) will be equivalent to Eq. (1). While it is possible to manually design weighting rules for setting $\boldsymbol{w}$ to promote label-discriminative learning, they will likely necessitate task-specific knowledge and nontrivial tuning. To facilitate the automatic learning of these weights $\boldsymbol{w}$, we propose to parameterize them as learnable hyperparameters using the idea of meta-learning.

**Meta Weight Learning Setup.** To automatically learn token weights using the idea of meta-learning, we formulate a bi-level optimization problem, where the inner objective $\mathcal{L}_{\text{w-gen}}$ optimizes the generator parameters $\boldsymbol{\theta}_p$, and the outer objective $\mathcal{L}_{\text{disc}}$ optimizes token weights $\boldsymbol{w}$ that are used as hyperparameters by the inner objective. We parameterize token weights $\boldsymbol{w}$ via a weighting network $g_{\boldsymbol{\omega}}$ so that $w_j = w_j(\boldsymbol{\omega})$. Details about the implementation of $g_{\boldsymbol{\omega}}$ are in Appendix E. Overall, the learning objectives are as follows:

$$\boldsymbol{\theta}_p^*(\boldsymbol{\omega}) = \arg\min_{\boldsymbol{\theta}_p} \mathcal{L}_{\text{w-gen}}, \quad \mathcal{L}_{\text{w-gen}}(\boldsymbol{\theta}_p; \boldsymbol{\omega}) = -\sum_{j=1}^{n} w_j(\boldsymbol{\omega}) \mathcal{L}_{\text{gen}}^j(\boldsymbol{\theta}_p)$$

$$\boldsymbol{\omega}^* = \arg\min_{\boldsymbol{\omega}} \mathcal{L}_{\text{disc}}, \quad \mathcal{L}_{\text{disc}}(\boldsymbol{\theta}_p^*(\boldsymbol{\omega})) = -\frac{1}{n}\sum_{j=1}^{n} \mathcal{L}_{\text{disc}}^j(\boldsymbol{\theta}_p^*(\boldsymbol{\omega})) \quad (4)$$

Under the above formulation, the token weights $w_j(\boldsymbol{\omega})$ are automatically learned such that the resulting generator parameters $\boldsymbol{\theta}_p^*(\boldsymbol{\omega})$ capture label-discriminative information (*i.e.*, minimize $\mathcal{L}_{\text{disc}}$). Instead of solving the optimal $\boldsymbol{\omega}^*$ and $\boldsymbol{\theta}_p^*$ via nested optimization loops, we use an online optimization strategy (Shu et al., 2019) for training efficiency. It also guarantees convergence to the critical points of both $\mathcal{L}_{\text{w-gen}}$ and $\mathcal{L}_{\text{disc}}$ under mild conditions. The initialization prompts can be found in Appendix C. The overall training procedure is shown in Algorithm 1.

**Analysis of Meta Weight Learning.** We analyze the gradient update of meta weights to study its effect in generator tuning. The weighting network parameter $\boldsymbol{\omega}$ is optimized via Eq. (4), and its

---

**Algorithm 2** Classification model fine-tuning on $\mathcal{D}_{\text{train}}$ and $\mathcal{D}_{\text{gen}}$.

---

**Input:** $\mathcal{D}_{\text{train}}$: Few-shot training set; $\mathcal{D}_{\text{gen}}$: Synthesized training set.
**Parameter:** $T$: Number of training steps.
**Output:** $\phi$: Trained classification model parameters.
$\phi^{(0)} \leftarrow$ Train on $\mathcal{D}_{\text{train}}$ with standard supervised learning
$\bar{z} \leftarrow \mathbf{0}$ `// Ensembled prediction initialization`
**for** $t \in [0, 1, \ldots, T-1]$ **do**
$\quad \mathcal{B} \leftarrow$ Sample a minibatch from $\mathcal{D}_{\text{gen}}$
$\quad \phi^{(t+1)} \leftarrow$ Take one gradient step to descend $\mathcal{L}_{\text{class}}$ in Eq. (5) on $\mathcal{B}$
$\quad \bar{z} \leftarrow$ Accumulate the current model prediction
$\quad$ Update $\mathcal{D}_{\text{gen}}$ to exclude noisy samples based on $\bar{z}$
**end**
**return** $\phi = \phi^{(T)}$

---

gradient is as follows (detailed derivation in Appendix A):

$$-\left.\frac{\partial \mathcal{L}_{\text{disc}}\left(\hat{\boldsymbol{\theta}}_p^{(t)}(\boldsymbol{\omega})\right)}{\partial \boldsymbol{\omega}}\right|_{\boldsymbol{\omega}=\boldsymbol{\omega}^{(t)}} \propto \sum_{j=1}^n d_j \left.\frac{\partial w_j(\boldsymbol{\omega})}{\partial \boldsymbol{\omega}}\right|_{\boldsymbol{\omega}=\boldsymbol{\omega}^{(t)}}, \quad d_j = \left.\frac{\partial \mathcal{L}_{\text{disc}}\left(\hat{\boldsymbol{\theta}}_p\right)}{\partial \hat{\boldsymbol{\theta}}_p}\right|_{\hat{\boldsymbol{\theta}}_p=\hat{\boldsymbol{\theta}}_p^{(t)}} \left.\frac{\partial \mathcal{L}_{\text{gen}}^j(\boldsymbol{\theta}_p)}{\partial \boldsymbol{\theta}_p}\right|_{\boldsymbol{\theta}_p=\boldsymbol{\theta}_p^{(t)}}^{\top}.$$

It can be seen that the gradient descent direction of $\boldsymbol{\omega}$ is determined by a weighted sum of token weight gradient ascent direction (*i.e.*, $\frac{\partial w_j(\boldsymbol{\omega})}{\partial \boldsymbol{\omega}}$), where the weight $d_j$ characterizes the similarity between the gradient of the discriminative objective and the gradient of the generative objective on the $j$th token. Therefore, the meta weights will be higher on those tokens where optimizing their generative objective is more beneficial for minimizing the discriminative objective.

### 3.3 CLASSIFIER FINE-TUNING

With the trained generator $G_{\boldsymbol{\theta}_p}$, we can synthesize novel training samples $\mathcal{D}_{\text{gen}}$ that augment $\mathcal{D}_{\text{train}}$ for fine-tuning a PLM $C_{\phi}$ for classification. The major challenge to effectively leverage $\mathcal{D}_{\text{gen}}$ is that the label noise (*i.e.*, some generated samples may not accurately pertain to the corresponding label) may deteriorate model performance if standard supervised learning is directly used. We propose a simple noise-robust training procedure to improve the generalization and stability of training: First fine-tune $C_{\phi}$ on $\mathcal{D}_{\text{train}}$ with standard supervised training, and then continue fine-tuning it on $\mathcal{D}_{\text{gen}}$ by applying *temporal ensembling* (Laine & Aila, 2017) as regularization. Specifically, given a training sample $(\tilde{\boldsymbol{x}}, \tilde{y}) \in \mathcal{D}_{\text{gen}}$, we minimize the following classification loss:

$$\min_{\phi} \mathcal{L}_{\text{class}}, \ \mathcal{L}_{\text{class}}(\phi) = -\log(p_{\phi}(\tilde{\boldsymbol{x}})_{\tilde{y}}) - \lambda \sum_{l=1}^L \bar{z}_l \log \frac{p_{\phi}(\tilde{\boldsymbol{x}})_l}{\bar{z}_l}, \tag{5}$$

where $p_{\phi}(\tilde{\boldsymbol{x}})$ is the model prediction on $\tilde{\boldsymbol{x}}$; $\lambda$ is a regularization weight for temporal ensembling; and $\bar{z}$ is the accumulated moving-average model predictions. We also use the ensembled prediction $\bar{z}$ to filter out noisy synthesized samples: We only include those samples for training where $\bar{z}$ strongly agrees with the label $\tilde{y}$ (*i.e.*, $\bar{z}_{\tilde{y}} > \delta$ where $\delta > 0$ is a threshold parameter). In Eq. (5), the first classification term is the cross-entropy loss; the second regularization term corresponds to temporal ensembling, which requires the current model prediction to be close to its past accumulated predictions. This not only neutralizes the fluctuation in model predictions for better training stability when label noise is present (Nguyen et al., 2020) but also helps prevent catastrophic forgetting (Kirkpatrick et al., 2017) of the information learned previously from the few-shot training set $\mathcal{D}_{\text{train}}$. Please refer to Appendix C for details about the temporal ensembling implementation. The overall procedure of classifier fine-tuning is summarized in Algorithm 2.

## 4 EXPERIMENTAL SETUP

**Downstream Tasks and Metrics.** We conduct evaluation on all tasks of the GLUE benchmark (Wang et al., 2018) (more details in Appendix B) except STS-B which is a regression task. We

Table 1: Results on seven classification tasks of the GLUE benchmark. We report average and standard deviation (as subscripts) performance over 5 different $\mathcal{D}_{train}/\mathcal{D}_{dev}$ splits defined in Gao et al. (2021). [†]: Results from Gao et al. (2021). [‡]: Results from Zhang et al. (2022). Methods that use additional models apart from the final classification model are marked.

| Method | MNLI-(m/mm) (Acc.) | QQP (F1) | QNLI (Acc.) | SST-2 (Acc.) | CoLA (Matt.) | RTE (Acc.) | MRPC (F1) | AVG |
|---|---|---|---|---|---|---|---|---|
| *Methods without Augmentation*: Few-shot samples are directly used for classifier tuning or as demonstrations for inference | | | | | | | | |
| Prompting[†] | 50.8/51.7 | 49.7 | 50.8 | 83.6 | 2.0 | 51.3 | 61.9 | 50.1 |
| Fine-Tuning[†] | $45.8_{6.4}/47.8_{6.8}$ | $60.7_{4.3}$ | $60.2_{6.5}$ | $81.4_{3.8}$ | $33.9_{14.3}$ | $54.4_{3.9}$ | $76.6_{2.5}$ | 59.1 |
| In-Context[†] | $52.0_{0.7}/53.4_{0.6}$ | $36.1_{5.2}$ | $53.8_{0.4}$ | $84.8_{1.3}$ | $-1.5_{2.4}$ | $60.4_{1.4}$ | $45.7_{6.0}$ | 47.4 |
| LM-BFF (Man.)[†] | $68.3_{2.3}/70.5_{1.9}$ | $65.5_{5.3}$ | $64.5_{4.2}$ | $92.7_{0.9}$ | $9.3_{7.3}$ | $69.1_{3.6}$ | $74.5_{5.3}$ | 63.6 |
| + demonstration[†] | $70.7_{1.3}/72.0_{1.2}$ | $69.8_{1.8}$ | $69.2_{1.9}$ | $92.6_{0.5}$ | $18.7_{8.8}$ | $68.7_{2.3}$ | $77.8_{2.0}$ | 66.9 |
| LM-BFF (Auto)[†] (w. 2.9B T5) | $68.3_{2.5}/70.1_{2.6}$ | $67.0_{3.0}$ | $68.3_{7.4}$ | $92.3_{1.0}$ | $14.0_{14.1}$ | $73.9_{2.2}$ | $76.2_{2.3}$ | 65.8 |
| + demonstration[†] (w. 2.9B T5) | $70.0_{3.6}/72.0_{3.1}$ | $67.7_{5.8}$ | $68.5_{5.4}$ | $93.0_{0.6}$ | $21.8_{15.9}$ | $71.1_{5.3}$ | $78.1_{3.4}$ | 67.3 |
| P-Tuning[‡] | $61.5_{2.1}/{-}$ | $65.6_{3.0}$ | $64.3_{2.8}$ | $92.2_{0.4}$ | — | — | $74.5_{7.6}$ | — |
| DART[‡] | $67.5_{2.6}/{-}$ | $67.8_{3.2}$ | $66.7_{3.7}$ | $93.5_{0.5}$ | — | — | $78.3_{4.5}$ | — |
| *Methods with Augmentation*: Few-shot samples are used for creating synthesized samples and for classifier tuning | | | | | | | | |
| MixText | $65.1_{2.6}/66.2_{2.8}$ | $60.6_{3.9}$ | $68.4_{5.1}$ | $89.1_{2.3}$ | $12.8_{9.2}$ | $66.5_{4.1}$ | $64.6_{7.6}$ | 61.1 |
| Back Translation (w. trained Marian) | $66.9_{4.6}/68.3_{3.8}$ | $59.8_{4.6}$ | $67.8_{4.9}$ | $91.1_{1.9}$ | $7.5_{3.7}$ | $62.4_{5.3}$ | $68.0_{11.2}$ | 60.6 |
| GPT3Mix (w. 175B GPT3) | $61.5_{3.2}/62.6_{2.2}$ | $70.4_{1.9}$ | $69.2_{0.3}$ | $\mathbf{93.6_{0.6}}$ | $\mathbf{48.9_{1.9}}$ | $70.4_{10.0}$ | $69.9_{12.4}$ | 69.2 |
| Generator Fine-Tuning (w. 1.6B CTRL) | $68.9_{5.1}/70.8_{5.3}$ | $60.4_{8.7}$ | $70.9_{4.1}$ | $91.2_{1.2}$ | $18.8_{10.0}$ | $66.1_{4.4}$ | $60.8_{15.4}$ | 62.6 |
| FewGen (w. 1.6B CTRL) | $\mathbf{75.7_{1.6}/77.1_{1.0}}$ | $\mathbf{71.5_{1.7}}$ | $\mathbf{76.3_{4.4}}$ | $93.1_{0.8}$ | $40.0_{7.5}$ | $71.2_{2.4}$ | $\mathbf{81.1_{2.5}}$ | **72.8** |
| Fully Supervised Fine-Tuning[†] | *89.8/89.5* | *81.7* | *93.3* | *95.0* | *62.6* | *80.9* | *91.4* | *84.9* |

follow the same data split and evaluation protocol as Gao et al. (2021): Both $\mathcal{D}_{train}$ and $\mathcal{D}_{dev}$ contain 16 samples per label and are sampled from the original training set with 5 different random seeds. The original development sets are used for testing. For all reported results, we include the average and standard deviation over the 5 different $\mathcal{D}_{train}/\mathcal{D}_{dev}$ splits. F1 score is used as the metric for QQP and MRPC, Matthews correlation for CoLA, and accuracy for the remaining tasks.

**Models, Training Settings and Hyperparameters.** FewGen is a training data generation method and can be used with any fine-tuning method on any classification model. We use moderate-sized PLMs to ensure our results are reproducible on typical research hardware: CTRL (1.6B parameters) (Keskar et al., 2019) as the generator $G_\theta$ and RoBERTa$_{Large}$ (356M parameters) (Liu et al., 2019) as the classifier $C_\phi$. We use prefix-tuning for training $G_\theta$ and prompt-based fine-tuning for training $C_\phi$. For simplicity, we use the most basic manual prompt version of LM-BFF (Gao et al., 2021). The only exception is CoLA for which we use the standard fine-tuning since the input data might be out of the distribution of $C_\phi$ (Gao et al., 2021). The hyperparameter tuning is performed on $\mathcal{D}_{dev}$. More details are in Appendix C.

**Compared Methods.** No-augmentation baselines include zero-shot prompting, standard fine-tuning, in-context learning, and the following strong few-shot learning methods: Four versions of LM-BFF (Gao et al., 2021), P-Tuning (Liu et al., 2021b) and DART (Zhang et al., 2022). We also compare FewGen with data augmentation methods for few-shot learning: MixText (Chen et al., 2020), using back translation systems to generate paraphrases (UDA-style (Xie et al., 2020) augmentation), GPT3Mix (Yoo et al., 2021) and standard fine-tuning of generator on the few-shot samples with prompts. All augmentation methods use LM-BFF (Man.) for fine-tuning the RoBERTa$_{Large}$ classifier. More details about data augmentation baselines can be found in Appendix D.

## 5 EVALUATION

### 5.1 MAIN RESULTS

We present the results of FewGen and baselines in Table 1. FewGen achieves overall better performance across the GLUE tasks, on average $5+$ points higher than the previous best few-shot method without augmentation, and $3+$ points better than GPT3Mix[2] (Yoo et al., 2021) which uses a 100 times larger generator model (175B) than FewGen. The promising results confirm the effectiveness of our

---

[2]The CoLA results reported in the original GPT3Mix paper use accuracy as the metric instead of Matthews correlation; our reimplemented GPT3Mix achieves $79.4_{0.6}$ on CoLA if measured by accuracy.

Table 2: Ablation studies by removing ($-$) or switching (w.) one component of FewGen.

| Method | MNLI-(m/mm) | QQP | QNLI | SST-2 | CoLA | RTE | MRPC |
|---|---|---|---|---|---|---|---|
| FewGen | $75.7_{1.6}/77.1_{1.0}$ | $71.5_{1.7}$ | $76.3_{4.4}$ | $93.1_{0.8}$ | $40.0_{7.5}$ | $71.2_{2.4}$ | $81.1_{2.5}$ |
| w. $\mathcal{L}_{\text{gen}}$ | $74.9_{1.0}/76.2_{1.0}$ | $70.7_{1.9}$ | $75.0_{4.8}$ | $92.5_{0.7}$ | $37.8_{8.2}$ | $69.5_{2.2}$ | $80.8_{3.0}$ |
| w. $\mathcal{L}_{\text{gen}} + \mathcal{L}_{\text{disc}}$ | $74.6_{1.6}/76.0_{1.5}$ | $68.8_{2.1}$ | $76.1_{4.3}$ | $92.4_{0.8}$ | $41.2_{9.0}$ | $70.1_{2.2}$ | $79.6_{2.4}$ |
| $-$ temporal ensemble | $72.2_{2.5}/74.0_{2.2}$ | $65.8_{2.1}$ | $75.1_{2.7}$ | $92.1_{1.7}$ | $33.9_{4.4}$ | $66.6_{2.4}$ | $80.4_{3.2}$ |
| w. fine-tune on $\mathcal{D}_{\text{train}} \cup \mathcal{D}_{\text{gen}}$ | $68.9_{1.8}/70.6_{1.9}$ | $64.3_{1.5}$ | $71.1_{4.1}$ | $91.8_{1.3}$ | $34.0_{3.2}$ | $59.6_{1.0}$ | $80.4_{3.5}$ |

proposed FewGen method in generating quality training data and leveraging them in combination with the few-shot training set for fine-tuning the classification model.

**Comparison with Back Translation.** Using back translation to paraphrase the few-shot samples does not improve the results, even with prompt-based fine-tuning to train the classifier – this is probably because it does not produce samples that are sufficiently different from the few-shot training set. The success of UDA (Xie et al., 2020) is grounded in the augmentations from abundant unlabeled data that improve the classifier generalization. However, under the strict few-shot learning setup, there is no access to additional task-specific unlabeled data (Gao et al., 2021), making it challenging for paraphrase-based methods to create sufficiently diverse training samples only based on the small few-shot set. The new training samples produced by our FewGen method are not limited to the paraphrases of the few-shot samples, as the generator is trained via prefix-tuning to preserve the PLM's pretraining knowledge, based on which novel training samples can be synthesized.

**Comparison with GPT3Mix.** The gigantic size of GPT3 makes it challenging for tuning on few-shot samples. Therefore, GPT3Mix (Yoo et al., 2021) uses few-shot samples as demonstrations for creating the augmentations. Such an approach suffers from two limitations: (1) Without any parameter update to the PLM, its learning ability is not fully leveraged to adapt to the few-shot training set. (2) The PLM can only use a small subset of the few-shot samples at a time for creating each augmentation, as the number of demonstrations received by the model is bounded by its maximum input sequence length. This makes the quality of the created augmentations more sensitive to the randomly drawn training samples. Our FewGen method, on the other hand, can use the entire few-shot set for tuning the PLM and achieves overall even better classification results with a much smaller PLM ($< 1\%$ the size of the GPT3 model) which can be deployed much more easily in practice.

## 5.2 ABLATION STUDIES

We further analyze the effectiveness of each important component in FewGen. Specifically, we compare FewGen with the following ablations: (1) Using the standard $\mathcal{L}_{\text{gen}}$ in Eq. (1) instead of our proposed $\mathcal{L}_{\text{w-gen}}$ in Eq. (3) for generator tuning (w. $\mathcal{L}_{\text{gen}}$); (2) using the directly combined $\mathcal{L}_{\text{gen}}$ and $\mathcal{L}_{\text{disc}}$ for generator tuning (w. $\mathcal{L}_{\text{gen}} + \mathcal{L}_{\text{disc}}$); (3) without applying temporal ensembling in Eq. (5) ($-$ temporal ensemble); (4) directly fine-tuning the classification model on the combination of $\mathcal{D}_{\text{gen}}$ and $\mathcal{D}_{\text{train}}$ (w. fine-tune on $\mathcal{D}_{\text{train}} \cup \mathcal{D}_{\text{gen}}$)[3]. As shown in Table 2, (1) & (2) using the standard maximum likelihood loss or the combination of generation and discrimination losses to tune the generator both yield lower-quality training data and lead to degraded classification performance; (3) not applying temporal ensembling for fine-tuning the classifier is more prone to label noise in the generated samples; (4) fine-tuning the classifier on the combination of $\mathcal{D}_{\text{gen}}$ and $\mathcal{D}_{\text{train}}$ significantly underperforms our two-step fine-tuning method. To study the impact of the amount of generated training samples on the model performance, we plot the MNLI-m accuracy (mean and standard deviation) with different sizes of $\mathcal{D}_{\text{gen}}$ in Fig. 3. Both the average model performance and stability improve with more generated samples.

## 5.3 ANALYSES OF LOSS FUNCTIONS FOR GENERATOR TUNING

As shown in Table 2, the choice of generator loss has a significant impact on the synthesized data quality and thus the final model performance. We conduct further analyses to compare the training processes of the generator under the following three loss functions and the resulting generated

---

[3]For this ablation, we upsample $\mathcal{D}_{\text{train}}$ by $\times 100$ so that its size is comparable with $\mathcal{D}_{\text{gen}}$; without upsampling, the result is much worse.

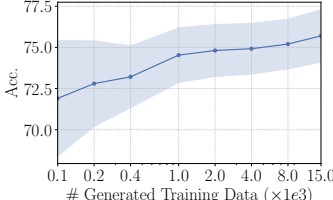

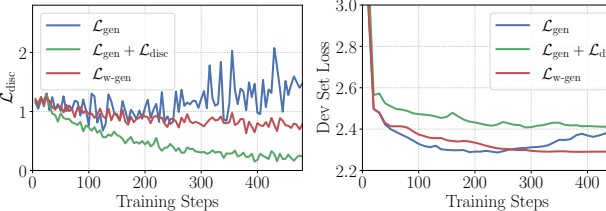

Figure 3: MNLI-m accuracy with different amounts of generated training data.

Figure 4: With different loss functions used for generator tuning, (Left) $\mathcal{L}_{\text{disc}}$ and (Right) standard language modeling loss on the development set. Best viewed in color.

samples: (1) $\mathcal{L}_{\text{gen}}$ which is the standard language modeling loss; (2) $\mathcal{L}_{\text{gen}} + \mathcal{L}_{\text{disc}}$ which directly adds the discriminative loss to generator training; and (3) $\mathcal{L}_{\text{w-gen}}$ which is our meta-weighted objective. Fig. 4 shows the discriminative loss $\mathcal{L}_{\text{disc}}$ and the standard language modeling loss on the held-out development set throughout training. Although using $\mathcal{L}_{\text{gen}} + \mathcal{L}_{\text{disc}}$ helps reduce the discriminative loss, it comes at the cost of hindering language modeling—the generator loss on the development set is high. Using our meta-weighted objective $\mathcal{L}_{\text{w-gen}}$ for tuning the generator not only encourages discriminativeness but also mitigates overfitting, yielding the lowest validation set loss. This is probably because the model receives contrastive information from other labels which facilitates more accurate modeling of the texts with the target label. We present more quantitative analyses of different generator training objectives in Appendix G. We visualize the token weights $w$ automatically learned and used in $\mathcal{L}_{\text{w-gen}}$ in Appendix F.

## 6 DISCUSSIONS AND CONCLUSIONS

**Ethical Considerations.** Despite the impressive text generation and representation power of PLMs, they can also come with the risk (Bender et al., 2021; Bender & Koller, 2020; Brown et al., 2020) of generating disinformation (Pagnoni et al., 2021) or exacerbating biases (Prabhumoye et al., 2018). Instead of improving upon PLM architectures or generation techniques, our work focuses on using existing PLMs to create training data for NLU tasks. Therefore, our method can be combined with any bias reduction and correction strategies (Gehman et al., 2020; Ma et al., 2020) in practice to reduce the adverse effects of PLMs.

**Limitations.** Compared to few-shot learning methods that directly train classification models on the small training set, FewGen requires tuning a generator PLM and using it to synthesize novel training samples, resulting in higher computation costs and longer running time. Still, we believe that our method may bring more good than harm—when the small training data size becomes the performance bottleneck for NLU tasks, a simple yet costly solution is to obtain more human annotations. Our method may replace or reduce the human efforts in such training data creation processes.

**Conclusions.** In this work, we propose FewGen, which leverages few-shot training samples to tune a generator PLM for synthesizing novel training data. The generated data can be then used in combination with few-shot samples to fine-tune a classification model for better generalization. To emphasize label-discriminative information during generator tuning, we propose a weighted maximum likelihood objective where the token weights are automatically learned via a discriminative meta objective. Since the generated samples may contain label noise, we propose a simple training procedure that first trains classifiers on the few-shot training set and then on the generated set by applying temporal ensembling for noise-robustness. Across seven classification tasks from the GLUE benchmark, FewGen significantly outperforms existing approaches under the same few-shot learning setting. The effectiveness of each important component in FewGen is validated via ablation studies. Future work directions may include: Using larger PLMs as the generator and the classifier, jointly training both models with each other's high-confident predictions, and developing systematic metrics for evaluating the quality of generated training samples.

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

## A   DERIVATION OF META WEIGHT GRADIENT UPDATE

We first write out the gradient update of $\hat{\boldsymbol{\theta}}_p^{(t)}\left(\boldsymbol{\omega}^{(t)}\right)$ and $\boldsymbol{\omega}^{(t+1)}$ according to Algorithm 1 as follows:

$$
\hat{\boldsymbol{\theta}}_p^{(t)}\left(\boldsymbol{\omega}^{(t)}\right) = \boldsymbol{\theta}_p^{(t)} - \alpha \left.\frac{\partial \mathcal{L}_{\text{w-gen}}\left(\boldsymbol{\theta}_p; \boldsymbol{\omega}^{(t)}\right)}{\partial \boldsymbol{\theta}_p}\right|_{\boldsymbol{\theta}_p = \boldsymbol{\theta}_p^{(t)}} = \boldsymbol{\theta}_p^{(t)} - \alpha \sum_{j=1}^n w_j\left(\boldsymbol{\omega}^{(t)}\right) \left.\frac{\partial \mathcal{L}_{\text{gen}}^j(\boldsymbol{\theta}_p)}{\partial \boldsymbol{\theta}_p}\right|_{\boldsymbol{\theta}_p = \boldsymbol{\theta}_p^{(t)}} \tag{6}
$$

$$
\boldsymbol{\omega}^{(t+1)} = \boldsymbol{\omega}^{(t)} - \beta \left.\frac{\partial \mathcal{L}_{\text{disc}}\left(\hat{\boldsymbol{\theta}}_p^{(t)}(\boldsymbol{\omega})\right)}{\partial \boldsymbol{\omega}}\right|_{\boldsymbol{\omega} = \boldsymbol{\omega}^{(t)}}. \tag{7}
$$

where $\alpha$ and $\beta$ are step sizes.

The gradient in Equation (7) is calculated as:

$$
\begin{aligned}
&\left.\frac{\partial \mathcal{L}_{\text{disc}}\left(\hat{\boldsymbol{\theta}}_p^{(t)}(\boldsymbol{\omega})\right)}{\partial \boldsymbol{\omega}}\right|_{\boldsymbol{\omega} = \boldsymbol{\omega}^{(t)}} \\
&= \left.\frac{\partial \mathcal{L}_{\text{disc}}\left(\hat{\boldsymbol{\theta}}_p\right)}{\partial \hat{\boldsymbol{\theta}}_p}\right|_{\hat{\boldsymbol{\theta}}_p = \hat{\boldsymbol{\theta}}_p^{(t)}} \left.\frac{\partial \hat{\boldsymbol{\theta}}_p(\boldsymbol{\omega})}{\partial \boldsymbol{\omega}}\right|_{\boldsymbol{\omega} = \boldsymbol{\omega}^{(t)}} \\
&= \left.\frac{\partial \mathcal{L}_{\text{disc}}\left(\hat{\boldsymbol{\theta}}_p\right)}{\partial \hat{\boldsymbol{\theta}}_p}\right|_{\hat{\boldsymbol{\theta}}_p = \hat{\boldsymbol{\theta}}_p^{(t)}} \left(-\alpha \sum_{j=1}^n \left.\frac{\partial \mathcal{L}_{\text{gen}}^j(\boldsymbol{\theta}_p)}{\partial \boldsymbol{\theta}_p}\right|_{\boldsymbol{\theta}_p = \boldsymbol{\theta}_p^{(t)}}^\top \left.\frac{\partial w_j(\boldsymbol{\omega})}{\partial \boldsymbol{\omega}}\right|_{\boldsymbol{\omega} = \boldsymbol{\omega}^{(t)}}\right) \qquad \text{Plugging in Eq. (6)} \\
&= -\alpha \sum_{j=1}^n \left( \underbrace{\left.\frac{\partial \mathcal{L}_{\text{disc}}\left(\hat{\boldsymbol{\theta}}_p\right)}{\partial \hat{\boldsymbol{\theta}}_p}\right|_{\hat{\boldsymbol{\theta}}_p = \hat{\boldsymbol{\theta}}_p^{(t)}} \left.\frac{\partial \mathcal{L}_{\text{gen}}^j(\boldsymbol{\theta}_p)}{\partial \boldsymbol{\theta}_p}\right|_{\boldsymbol{\theta}_p = \boldsymbol{\theta}_p^{(t)}}^\top}_{\triangleq d_j} \left.\frac{\partial w_j(\boldsymbol{\omega})}{\partial \boldsymbol{\omega}}\right|_{\boldsymbol{\omega} = \boldsymbol{\omega}^{(t)}} \right)
\end{aligned}
$$

Therefore,

$$
-\left.\frac{\partial \mathcal{L}_{\text{disc}}\left(\hat{\boldsymbol{\theta}}_p^{(t)}(\boldsymbol{\omega})\right)}{\partial \boldsymbol{\omega}}\right|_{\boldsymbol{\omega} = \boldsymbol{\omega}^{(t)}} \propto \sum_{j=1}^n d_j \left.\frac{\partial w_j(\boldsymbol{\omega})}{\partial \boldsymbol{\omega}}\right|_{\boldsymbol{\omega} = \boldsymbol{\omega}^{(t)}}, \quad d_j = \left.\frac{\partial \mathcal{L}_{\text{disc}}\left(\hat{\boldsymbol{\theta}}_p\right)}{\partial \hat{\boldsymbol{\theta}}_p}\right|_{\hat{\boldsymbol{\theta}}_p = \hat{\boldsymbol{\theta}}_p^{(t)}} \left.\frac{\partial \mathcal{L}_{\text{gen}}^j(\boldsymbol{\theta}_p)}{\partial \boldsymbol{\theta}_p}\right|_{\boldsymbol{\theta}_p = \boldsymbol{\theta}_p^{(t)}}^\top
$$

## B   GLUE TASKS

We provide the details of the seven classification tasks included in the GLUE benchmark.

**MNLI:** Multi-genre Natural Language Inference (Williams et al., 2018) requires predicting whether a given premise sentence entails, contradicts or neutral with respect to a given hypothesis sentence.

**QQP:** Quora Question Pairs (Shankar et al., 2017) requires judging whether a pair of questions asked are semantically equivalent.

**QNLI:** Question Natural Language Inference requires predicting whether a given sentence contains the answer to a given question sentence.

**SST-2:** Stanford Sentiment Treebank (Socher et al., 2013) requires determining if a movie review has positive or negative sentiment.

**CoLA:** Corpus of Linguistic Acceptability (Warstadt et al., 2019) requires determining whether a given sentence is linguistically acceptable or not.

**RTE:** Recognizing Textual Entailment (Bentivogli et al., 2009; Dagan et al., 2005; Giampiccolo et al., 2007; Haim et al., 2006) requires predicting whether a given premise sentence entails a given hypothesis sentence or not.

**MRPC:** Microsoft Research Paraphrase Corpus (Dolan & Brockett, 2005) requires predicting whether two sentences are semantically equivalent or not.

## C  IMPLEMENTATION DETAILS

Table 3: Prompts used for initializing the prefix vectors and control codes (required by CTRL (Keskar et al., 2019)) used in generator training. The control codes are selected to approximate the domain of the task. For single-sequence tasks, $x$ denotes the training sample; for sequence-pair tasks, $x_1$ and $x_2$ denote the first and second sequence in the training sample, respectively.

| Task | Task Type | Control Code | Label | Initialization Prompt |
|---|---|---|---|---|
| **SST-2** | single-sequence | Reviews | positive
negative | Rating: 5.0 positive movie review: $x$
Rating: 1.0 negative movie review: $x$ |
| **CoLA** | single-sequence | Links | grammatical
not grammatical | Linguistically correct sentence: $x$
Linguistically incorrect sentence: $x$ |
| **MNLI** | sequence-pair | Wikipedia | entailment
neutral
contradiction | Sentence 1 implies Sentence 2. Sentence 1: $x_1$ Sentence 2: $x_2$
Sentence 2 supplements Sentence 1. Sentence 1: $x_1$ Sentence 2: $x_2$
Sentence 2 contradicts Sentence 1. Sentence 1: $x_1$ Sentence 2: $x_2$ |
| **QNLI** | sequence-pair | Links | entailment
not entailment | Paragraph is relevant to Question. Question: $x_1$ Paragraph: $x_2$
Paragraph is irrelevant to Question. Question: $x_1$ Paragraph: $x_2$ |
| **RTE** | sequence-pair | Wikipedia | entailment
not entailment | Sentence 1 implies Sentence 2. Sentence 1: $x_1$ Sentence 2: $x_2$
Sentence 2 supplements Sentence 1. Sentence 1: $x_1$ Sentence 2: $x_2$ |
| **MRPC** | sequence-pair | Wikipedia | equivalent
not equivalent | Sentence 1 is equivalent to Sentence 2. Sentence 1: $x_1$ Sentence 2: $x_2$
Sentence 1 is different from Sentence 2. Sentence 1: $x_1$ Sentence 2: $x_2$ |
| **QQP** | sequence-pair | Links | equivalent
not equivalent | Question 1 is equivalent to Question 2. Question 1: $x_1$ Question 2: $x_2$
Question 1 is different from Question 2. Question 1: $x_1$ Question 2: $x_2$ |

**Details of Initialization Prompts Used for Generator Tuning on Different Tasks.** For generator tuning, we find it beneficial to initialize the prefix vectors with task-descriptive prompts, similar to the observations in Li & Liang (2021). The prefix lengths (*i.e.*, number of trained prefix token positions) are equal to the number of tokens in the prompts. We present details about the prompts used for initializing the prefix vectors for different tasks in Table 3. For sequence-pair tasks, an additional infix prompt is used between the two sequences, and we also tune the embeddings of the infix (*i.e.*, prompt-tuning (Lester et al., 2021)) for generator training.

**Details of Generator Tuning.** The meta-weighted generator tuning procedure (Algorithm 1) involves three forward and backward passes, and thus its time complexity is approximately 3 times of standard generator training without meta learning. However, since the few-shot training sets have a small amount of training data, the extra time cost is usually affordable. In practice, our generator tuning with meta weight learning takes 10 minutes to train on each task (the standard generator training time without meta-learning is 3.5 minutes). We use a fixed set of hyperparamters for all tasks without task-specific hyperparamter tuning: We set batch size to be 2, the learning rate for optimizing $\theta_p$ to be $5e-3$, the learning rate for optimizing $\omega$ to be $1e-2$, and training epoch to be 20.

**Details of Generating Training Data.** Following Meng et al. (2022), for sequence-pair tasks (MNLI, QQP, QNLI, RTE and MRPC), we randomly sample the first sequence from the pretraining corpus (*e.g.*, Wikipedia) and use greedy sampling for generating the second sequence. For single-sequence tasks (SST-2 and CoLA), we use top-$k$ sampling with temperature to generate training data from scratch where $k = 10$. For all tasks, we generate $5,000$ samples per label.

For SST-2, we use one of the following tokens to start generation: "a", "one", "the", "this", "that", "i", "you", "it", "what". For CoLA, we use a random stop word to start generation.

**Hyperparameters for Fine-Tuning Classifier PLMs.** For fine-tuning on the few-shot training samples $\mathcal{D}_{\text{train}}$, we search among the following hyperparameter ranges based on development set ($\mathcal{D}_{\text{dev}}$) model performance and pick the best performing model for futher fine-tuning on synthesized data: Learning rate in $[1e-5, 2e-5]$ and batch size in $[2, 4, 8]$. The number of training steps is fixed to be $1000$. We also find it beneficial to apply label smoothing (smoothing weight set to $0.15$) for fine-tuning on the few-shot training set.

For fine-tuning on the synthesized training samples $\mathcal{D}_{\text{gen}}$, we use the following hyperparameters: $5e - 6$ as the learning rate; 16 as the batch size; label smoothing weight $\epsilon = 0.15$ ; temporal ensemble momentum $\gamma = 0.9$; temporal ensemble loss weight $\lambda = 20$; training steps $T = 6,000$.

**Details of Temporal Ensembling for Fine-Tuning Classifier PLMs on Synthetic Data.** We update ensembled predictions $\bar{z}$ as follows where $p_\phi$ is the current model prediction, $\gamma$ is the momentum parameter, $\hat{z}$ is the accumulated model prediction before bias correction, $\bar{z}$ is the accumulated model prediction after bias correction, and $t$ is the number of updates $\bar{z}$ has received:

$$\hat{z} \leftarrow \gamma\hat{z} + (1 - \gamma)p_\phi, \ \bar{z} \leftarrow \hat{z}/(1 - \gamma^t).$$

The accumulated model prediction $\hat{z}$ has a zero initialization; the division $(1 - \gamma^t)$ is for bias correction (Laine & Aila, 2017). After each update of $\hat{z}$, it will be compared to a threshold value $\delta$; each synthesized sample $(\tilde{x}, \tilde{y})$ will be included in training only if $\bar{z}_{\tilde{y}} > \delta$.

We update the ensembled predictions $\bar{z}$ on all samples in $\mathcal{D}_{\text{gen}}$ every 200 steps, and set the threshold value for sample filtering $\delta = 0.8$.

**Computation Environment.** The experiments are conducted on NVIDIA A100 GPUs.

## D  DATA AUGMENTATION BASELINE DETAILS

**Details About MixText (Chen et al., 2020).** We use the TMix version of MixText to perform data interpolation on the few-shot labeled dataset (since there is no access to unlabeled task-specific data under the strict few-shot learning setting Gao et al. (2021)). We adapt the label mix-up operation to fit prompt-based fine-tuning by interpolating the label words instead of categorical labels; we observe that this results in better few-shot performance than the original TMix, probably analogous to why prompt-based fine-tuning outperforms standard fine-tuning for few-shot learning. We train the classifier with supervised loss combined with consistency loss over the interpolated samples as in the original paper. We follow the default hyperparameters in MixText.

**Details About Back Translation.** We use two trained Marian (Junczys-Dowmunt et al., 2018) models to perform data augmentation via back translation. We translate our labeled examples from English to French, and then back to English. As in UDA (Xie et al., 2020), we employ random sampling with a tunable temperature to generate a diverse set of derivative examples. We generate 32 examples from each few-shot training example and let the synthesized samples share the same label with the original few-shot training sample. After combining with the original examples, we fine-tune the classifier and observe performance.

**Details About GPT3Mix (Yoo et al., 2021).** We use the 175B GPT3 model for generating the augmentations. For creating each augmentation, we randomly sample $k = 4$ (the optimal setting according to GPT3Mix) examples from the few-shot training set as demonstrations. The prompts follow the suggested format proposed in the original paper (Yoo et al., 2021) and are shown in Table 4. We create $5,000$ augmented samples per label to make the resulting training set size equal to that of FewGen. After obtaining the augmented examples and their pseudo labels (the probability predictions over all labels by GPT3), we use them along with the real few-shot samples for fine-tuning the classifier, following the setting in GPT3Mix (Yoo et al., 2021).

**Details About Standard Generator Fine-Tuning.** We fine-tune the same 1.6B CTRL (Keskar et al., 2019) model as used in FewGen with the standard maximum likelihood objective. Different from previous studies (Anaby-Tavor et al., 2020; Kumar et al., 2020) that prepend categorical labels to the training samples, we enhance the generator fine-tuning with label-descriptive prompts (shown in Table 3) used in FewGen. We create $5,000$ augmented samples per label to make the resulting training set size equal to that of FewGen.

## E  DETAILS OF WEIGHTING NETWORK IMPLEMENTATION

Table 4: Prompts used for GPT3Mix augmentation. For sequence-pair tasks, $x_1$ and $x_2$ denote the first and second input sequence, respectively. For single-sequence tasks, $x$ denotes the input sequence. $y$ denotes the label name. Only one example is shown in the template for clarity; in practice, we concatenate $k = 4$ samples according to the optimal setting in GPT3Mix (Yoo et al., 2021).

| Task | Template | Label name |
|------|----------|-----------|
| SST-2 | Each item in the following list contains a movie review and the respective sentiment. The sentiment is one of 'positive' or 'negative'. Movie review: $x$ (Sentiment: $y$) ... | positive: positive
negative: negative |
| CoLA | Each item in the following list contains a text and the respective grammar. The grammar is one of 'correct' or 'incorrect'. Text: $x$ (Grammar: $y$) ... | grammatical: correct
not grammatical: incorrect |
| MNLI | Each item in the following list contains a premise, a hypothesis and their logical relation. The logical relation is one of 'entailment', 'neutral' or 'contradiction'. Premise: $x_1$ Hypothesis: $x_2$ (Logical relation: $y$) ... | entailment: entailment
neutral: neutral
contradiction: contradiction |
| QNLI | Each item in the following list contains a question, an answer and their logical relation. The logical relation is one of 'entailment' or 'neutral'. Question: $x_1$ Answer: $x_2$ (Logical relation: $y$) ... | entailment: entailment
not entailment: neutral |
| RTE | Each item in the following list contains a premise, a hypothesis and their logical relation. The logical relation is one of 'entailment' or 'neutral'. Premise: $x_1$ Hypothesis: $x_2$ (Logical relation: $y$) ... | entailment: entailment
not entailment: neutral |
| MRPC | Each item in the following list contains two sentences and their semantic relation. The semantic relation is one of 'equivalent' or 'different'. Sentence 1: $x_1$ Sentence 2: $x_2$ (Semantic relation: $y$) ... | equivalent: equivalent
not equivalent: different |
| QQP | Each item in the following list contains two questions and their semantic relation. The semantic relation is one of 'equivalent' or 'different'. Question 1: $x_1$ Question 2: $x_2$ (Semantic relation: $y$) ... | equivalent: equivalent
not equivalent: different |

Since the token weights $w$ used in Eq. (4) need to characterize the discriminativeness of each token, we use the value of discriminative objective at each token $\mathcal{L}_{\text{disc}}^j$ as the input to the weighting network, and we use softmax to normalize the weights:

$$w_j(\boldsymbol{\omega}) = \frac{\exp\left(g_{\boldsymbol{\omega}}(\mathcal{L}_{\text{disc}}^j)\right)}{\sum_{j'=1}^n \exp\left(g_{\boldsymbol{\omega}}(\mathcal{L}_{\text{disc}}^{j'})\right)}.$$

Following Shu et al. (2019), we instantiate $g_{\boldsymbol{\omega}}$ to be a feedforward network (FFN) with only one 100-dimension hidden layer by default. We

Table 5: Study of weighting network instantiation. The default architecture is a feedforward network (FFN) with one hidden layer. We also explore adding a self-attention layer on top of the generator PLM's output hidden states (Self-attention). We use the same two metrics with Table 6 to evaluate the resulting generators.

| Architecture | MNLI | | SST-2 | |
|---|---|---|---|---|
| | Acc. (↑) | PPL (↓) | Acc. (↑) | PPL (↓) |
| FFN | **72.3** | **11.9** | **93.2** | **43.5** |
| Self-attention | 70.3 | 12.9 | 92.3 | 44.2 |

explore an alternative instantiation that adds one self-attention layer on top of the generator PLM's output hidden states. The meta weights are finally obtained by projecting the outputs of the self-attention layer using another linear layer. We evaluate the resulting generator quality via the same two metrics as in Appendix G. Table 5 shows that using more complicated architectures (*e.g.*, adding another self-attention layer) does not result in a better generator compared to using a simple FFN for meta weight learning. This is probably because the generator PLM's output representations are sufficiently contextualized and contain the information necessary for learning the token weights, thus a simple FFN as the weighting network will be enough. Using more complicated networks, on the other hand, will introduce more randomly initialized new parameters which may not be learned well using the limited amount of few-shot training data.

## F  VISUALIZATION OF TOKEN WEIGHT LEARNING

To gain intuitive understanding of what tokens are assigned more weight during generator tuning, we visualize the learned weights in Fig. 5. The tokens with higher weights (*e.g.*, "weak" in the first example and "hates" in the second example) are learned to be important tokens that decide the relation of the second sentence to the first sentence (*i.e.*, the label of the training sample). With such tokens emphasized during training, the generator is encouraged to capture label-discriminative information that facilitates the generation of unambiguous training samples.

**Sentence 1: But prophecy is always strongest when based on coincidence--that is a prime rule.**
**Sentence 2: Prophecies based on coincidences are widely known to be weak and unreliable.**
**Label: Contradiction**

weights     0.03     0.02   0.03     0.11      0.06   0.06    0.03 0.03 0.05 0.33 0.06     0.21

**Sentence 1: But Rodgers did tell Lewis that he despises Amelio because Amelio supported**
**Clinton, so it is Rodgers' mistake, not our author's, that we are correcting.**
**Label: Entailment**
**Sentence 2: Rodgers told Lewis he hates Amelio.**

weights     0.14    0.08    0.07   0.08   0.47    0.17

Figure 5: Visualization of learned token weights on two samples from MNLI's few-shot training set. The generator is trained given the first sentence to generate the second. The tokens associated with higher weights are more label indicative.

Table 6: Evaluation of generator training objectives. We use two metrics: Generated data accuracy (Acc; higher is better) and generator's perplexity on the test set (PPL; lower is better). The results are averaged over 5 $\mathcal{D}_{\text{train}}/\mathcal{D}_{\text{dev}}$ splits.

| Objective | MNLI | | QQP | | QNLI | | SST-2 | | CoLA | | RTE | | MRPC | |
|---|---|---|---|---|---|---|---|---|---|---|---|---|---|---|
| | Acc. (↑) | PPL (↓) | Acc. (↑) | PPL (↓) | Acc. (↑) | PPL (↓) | Acc. (↑) | PPL (↓) | Acc. (↑) | PPL (↓) | Acc. (↑) | PPL (↓) | Acc. (↑) | PPL (↓) |
| $\mathcal{L}_{\text{gen}}$ | 69.4 | 13.1 | 87.5 | 10.9 | 57.0 | 23.4 | 91.5 | 43.8 | 59.1 | 85.6 | 82.9 | 9.3 | 87.6 | 5.0 |
| $\mathcal{L}_{\text{gen}} + \mathcal{L}_{\text{disc}}$ | 70.2 | 13.5 | 87.3 | 11.2 | 57.2 | 24.8 | 92.0 | 49.5 | 59.2 | 87.0 | 82.8 | 9.6 | 86.3 | 5.3 |
| $\mathcal{L}_{\text{w-gen}}$ | **72.3** | **11.9** | **89.5** | **10.7** | **60.1** | **23.2** | **93.2** | **43.5** | **60.7** | **83.8** | **83.4** | **8.9** | **90.5** | **4.9** |

Table 7: (For MNLI) Examples of generated second sequence (hypothesis) by generators tuned with three different objectives conditioned on a given first sequence (premise) "*In 2009, hair samples from 1,137 Taiji residents were tested for mercury by the National Institute for Minamata Disease (NIMD)*". The true label of the generated sequence is marked at the end of the sequence (if the generated sequence correctly pertains to the target label, it is marked in blue; otherwise, it is in red).

| Objective | Label | Generated Second Sequence |
|---|---|---|
| $\mathcal{L}_{\text{gen}}$ | entailment | The NIMD found that hair samples from 1,137 Taiji residents had mercury in their hair. *(neutral)* |
| | neutral | The NIMD found that there was no evidence of a link between exposure to high levels of mercury and thyroid cancer. *(neutral)* |
| | contradiction | There was no evidence of mercury in hair samples from Taiji. *(neutral)* |
| $\mathcal{L}_{\text{gen}} + \mathcal{L}_{\text{disc}}$ | entailment | The number of hairs in a sample is equal to the number of people who lived in Taiji. *(neutral)* |
| | neutral | The results showed that there was no significant difference in levels of mercury. *(neutral)* |
| | contradiction | Hair samples from 1,137 Taiji residents were not tested. *(contradiction)* |
| $\mathcal{L}_{\text{w-gen}}$ | entailment | The NIMD tested hair samples from 1,137 residents of Taiji. *(entailment)* |
| | neutral | There was no significant difference in levels between people who lived near a nickel mine and those living far away. *(neutral)* |
| | contradiction | The NIMD did not test any of the hair samples. *(contradiction)* |

# G QUANTITATIVE ANALYSES OF GENERATOR TRAINING OBJECTIVES

Apart from the final classification model performance which indirectly reflects the synthetic data quality, we additionally conduct more direct quantitative analyses of different generator training objectives. We use two metrics: (1) The accuracy of generated texts, which is judged by fully-supervised RoBERTa$_{\text{Large}}$ models fine-tuned on the original training sets of each task. We choose to adopt such an automatic evaluation instead of human evaluation because it is efficient and reliable—fully-supervised RoBERTa$_{\text{Large}}$ models have comparable or better accuracy than human baselines according to the GLUE benchmark[4]. (2) The generator's perplexity on the test sets, which reflects how well the generator models the task distribution. As shown in Table 6, using $\mathcal{L}_{\text{w-gen}}$ for generator training consistently outperforms using $\mathcal{L}_{\text{gen}}$ or $\mathcal{L}_{\text{gen}} + \mathcal{L}_{\text{disc}}$, both in generated text accuracy and in language modeling ability.

Comparing $\mathcal{L}_{\text{w-gen}}$ with $\mathcal{L}_{\text{gen}}$, the meta weights automatically learned emphasize discriminative tokens in generator training and help the generator capture the subtle semantic differences across different labels, resulting in better language modeling quality and more distinctive generated data.

Comparing $\mathcal{L}_{\text{w-gen}}$ with $\mathcal{L}_{\text{gen}} + \mathcal{L}_{\text{disc}}$, the generator training objective is not directly impacted by the discriminative objective, thus avoiding the gradient interference issue in multi-task learning (Standley et al., 2019)—the gradient for optimizing the generative probability $p(\boldsymbol{x}|y_l)$ will be interfered by

---

[4] https://gluebenchmark.com/leaderboard

that optimizing the discriminative probability $p(y_l|\boldsymbol{x})$ if $\mathcal{L}_{\text{gen}} + \mathcal{L}_{\text{disc}}$ is used. Therefore, using $\mathcal{L}_{\text{w-gen}}$ results in better language modeling quality and more fluent and coherent generation results.

We also showcase concrete generation results for the three labels of MNLI by models trained with the three different loss functions in Table 7. The model trained with $\mathcal{L}_{\text{gen}}$ produces fluent and coherent sentences, but the generated sentences do not accurately pertain to the desired label (*i.e.*, the "entailment" and "contradiction" generation results are in fact neutral with respect to the given sentence), lacking label discriminativeness. When $\mathcal{L}_{\text{gen}} + \mathcal{L}_{\text{disc}}$ is used, the generated samples of different labels are more distinctive, but also become less natural and coherent due to the model's language modeling ability being hampered. The generator tuned with $\mathcal{L}_{\text{w-gen}}$ produces both coherent and label-discriminative samples which can serve as quality training data.

Table 8: 16-shot training samples of SST-2.

| Label | Example | Review Text |
|---|---|---|
| positive | #1 | (ramsay) visually transforms the dreary expanse of dead-end distaste the characters inhabit into a poem of art , music and metaphor . |
| | #2 | the film jolts the laughs from the audience – as if by cattle prod . |
| | #3 | the film presents visceral and dangerously honest revelations about the men and machines behind the curtains of our planet . |
| | #4 | a film that will enthrall the whole family . |
| | #5 | serious movie-goers embarking upon this journey will find that the road to perdition leads to a satisfying destination . |
| | #6 | sweet and memorable film . |
| | #7 | shyamalan takes a potentially trite and overused concept (aliens come to earth) and infuses it into a rustic , realistic , and altogether creepy tale of hidden invasion . |
| | #8 | a crisp psychological drama (and) a fascinating little thriller that would have been perfect for an old " twilight zone " episode . |
| | #9 | my big fat greek wedding is not only the best date movie of the year , it 's also a – dare i say it twice – delightfully charming – and totally american , i might add – slice of comedic bliss . |
| | #10 | a comedy-drama of nearly epic proportions rooted in a sincere performance by the title character undergoing midlife crisis . |
| | #11 | diggs and lathan are among the chief reasons brown sugar is such a sweet and sexy film . |
| | #12 | you 're not merely watching history , you 're engulfed by it . |
| | #13 | the concept is a hoot . |
| | #14 | the filmmakers ' eye for detail and the high standards of performance convey a strong sense of the girls ' environment . |
| | #15 | a haunting tale of murder and mayhem . |
| | #16 | neil burger here succeeded in ... making the mystery of four decades back the springboard for a more immediate mystery in the present . |
| negative | #1 | nothing happens , and it happens to flat characters . |
| | #2 | as lively an account as seinfeld is deadpan . |
| | #3 | so we got ten little indians meets friday the 13th by way of clean and sober , filmed on the set of carpenter 's the thing and loaded with actors you 're most likely to find on the next inevitable incarnation of the love boat . |
| | #4 | the plot is nothing but boilerplate cliches from start to finish , and the script assumes that not only would subtlety be lost on the target audience , but that it 's also too stupid to realize that they 've already seen this exact same movie a hundred times |
| | #5 | ultimately , sarah 's dedication to finding her husband seems more psychotic than romantic , and nothing in the movie makes a convincing case that one woman 's broken heart outweighs all the loss we witness . |
| | #6 | the big finish is a bit like getting all excited about a chocolate eclair and then biting into it and finding the filling missing . |
| | #7 | this picture is mostly a lump of run-of-the-mill profanity sprinkled with a few remarks so geared toward engendering audience sympathy that you might think he was running for office – or trying to win over a probation officer . |
| | #8 | just because a walk to remember is shrewd enough to activate girlish tear ducts does n't mean it 's good enough for our girls . |
| | #9 | often lingers just as long on the irrelevant as on the engaging , which gradually turns what time is it there ? |
| | #10 | this movie , a certain scene in particular , brought me uncomfortably close to losing my lunch . |
| | #11 | but it would be better to wait for the video . |
| | #12 | a rude black comedy about the catalytic effect a holy fool has upon those around him in the cutthroat world of children 's television . |
| | #13 | just a collection of this and that – whatever fills time – with no unified whole . |
| | #14 | although god is great addresses interesting matters of identity and heritage , it 's hard to shake the feeling that it was intended to be a different kind of film . |
| | #15 | the chocolate factory without charlie . |
| | #16 | in that setting , their struggle is simply too ludicrous and borderline insulting . |

## H   CONCRETE GENERATION RESULTS

We present few-shot training samples ($\mathcal{D}_{\text{train}}$) of SST-2 in Table 8, and some concrete generation results (from $\mathcal{D}_{\text{gen}}$) for all tasks in Tables 9, 10, 11, 12, 13, 14, and 15.

As shown in Tables 8 and 9, the synthetic samples are accurate and quite different from the given training samples to serve as effective augmentations. For sequence-pair tasks, because we randomly sample the first sequence from the pretraining corpus and let the generator create the second sequence given certain labels, the resulting generated samples will be certainly different from the given training samples.

Table 9: Examples of concrete generations for SST-2.

| Example | Label | Generated Text |
|---|---|---|
| #1 | positive | i was a bit skeptical when i first saw the trailer for this movie, but after watching it, it was quite enjoyable. |
| | negative | i was expecting a fun and entertaining movie with some nice special effects, but instead it was just plain boring. |
| #2 | positive | that is, a film that has the potential to move an audience to tears, or at least to a state of mind where you feel something. |
| | negative | a film that has the potential to be one of the most moving and powerful experiences ever made is instead marred by an unfocused script, bad acting and a lack of emotional depth. |
| #3 | positive | this is a good movie – it's not just for people who like to see things blow up or have explosions or shoot guns. |
| | negative | i am a big fan of the first two movies but this one is just plain bad. |

Table 10: Examples of concrete generations for CoLA.

| Example | Label | Generated Text |
|---|---|---|
| #1 | grammatical | Between the two of us, I think you have a good chance to make it to the finals. |
| | not grammatical | A few weeks later we saw them on television. We thought there were a hundred more. It wasn't that they came out one at time. |
| #2 | grammatical | Doing the dishes is a task that I have to do because I am not allowed to go out and play with my friends. |
| | not grammatical | Only those whose lives depend on it, but only when you are at its mercy do it harm. |
| #3 | grammatical | To be able to speak the language of the other person is a prerequisite for communication. |
| | not grammatical | The only good way to avoid getting into trouble for being too friendly would to let me go on a trip. |

Table 11: Examples of concrete generations for QQP. The first question (*italicized*) is randomly sampled from the pretraining corpus; the second question (underlined) is generated by $G_{\boldsymbol{\theta}_p}$.

| Example | Label | Generated Text |
|---|---|---|
| #1 | equivalent | Question 1: *How long does it take for a project to be completed?*
Question 2: How long does it take to complete a project? |
| | not equivalent | Question 1: *How long does it take for a project to be completed?*
Question 2: What is the total cost of completing this project? |
| #2 | equivalent | Question 1: *What mascots or characters would you like to see included as Super Mario Maker DLC in the future?*
Question 2: What would you like to see in Super Mario Maker DLC that you did not see in the game? |
| | not equivalent | Question 1: *What mascots or characters would you like to see included as Super Mario Maker DLC in the future?*
Question 2: How do I get a copy of this game? |

Table 12: Examples of concrete generations for MNLI. The first sentence (*italicized*) is randomly sampled from the pretraining corpus; the second sentence (underlined) is generated by $G_{\theta_p}$.

| Example | Label | Generated Text |
|---|---|---|
| #1 | entailment | Sentence 1: *Air is provided for the combustion by an electric blower.*
Sentence 2: The blower provides air to a combustion chamber. |
| | neutral | Sentence 1: *Air is provided for the combustion by an electric blower.*
Sentence 2: Electric blowers are available in most gas stations. |
| | contradiction | Sentence 1: *Air is provided for the combustion by an electric blower.*
Sentence 2: The blower does not work. |
| #2 | entailment | Sentence 1: *Since its base is almost at sea level, it is only the 15th highest light in the United States, the first 14 being built on higher ground.*
Sentence 2: It is the 15th highest light in the United States. |
| | neutral | Sentence 1: *Since its base is almost at sea level, it is only the 15th highest light in the United States, the first 14 being built on higher ground.*
Sentence 2: The lighthouse was originally constructed to be a beacon for ships passing by and as such has been used since before World War II. |
| | contradiction | Sentence 1: *Since its base is almost at sea level, it is only the 15th highest light in the United States, the first 14 being built on higher ground.*
Sentence 2: It is located on a mountain top. |

Table 13: Examples of concrete generations for QNLI. The question (*italicized*) is randomly sampled from the pretraining corpus; the answer (underlined) is generated by $G_{\theta_p}$.

| Example | Label | Generated Text |
|---|---|---|
| #1 | entailment | Question: *What makes you want to step up to the next level?*
Answer: I want to be the best player I can be. |
| | not entailment | Question: *What makes you want to step up to the next level?*
Answer: The new program will be called "Project 10" and it is expected that a total of $450 million in federal funding would go toward it. |
| #2 | entailment | Question: *How do all those shops know what you would like to buy?*
Answer: The stores are able to track your preferences and provide you with a list of products that are best for you. |
| | not entailment | Question: *How do all those shops know what you would like to buy?*
Answer: The stores are not required by law or regulation in the United States and Canada but they have been known for years as a source of illegal sales on eBay. |

Table 14: Examples of concrete generations for RTE. The first sentence (*italicized*) is randomly sampled from the pretraining corpus; the second sentence (underlined) is generated by $G_{\theta_p}$.

| Example | Label | Generated Text |
|---|---|---|
| #1 | entailment | Sentence 1: *In 1970, the Science Council of Canada recommended that the government of Canada immediately invest in industrial development of the aviation industry, including construction of aircraft, navigation aids, and regulation of air traffic.*
Sentence 2: The government of Canada has invested in the aviation industry. |
| | not entailment | Sentence 1: *In 1970, the Science Council of Canada recommended that the government of Canada immediately invest in industrial development of the aviation industry, including construction of aircraft, navigation aids, and regulation of air traffic.*
Sentence 2: The Aviation Industry was established by a Royal Decree on June 1, 1970. |
| #2 | entailment | Sentence 1: *All of the Centre's staff are fluently bilingual in both English and Chinese and are familiar with the traditions of the Chinese culture.*
Sentence 2: The Centre is a bilingual institution. |
| | not entailment | Sentence 1: *All of the Centre's staff are fluently bilingual in both English and Chinese and are familiar with the traditions of the Chinese culture.*
Sentence 2: The Centre is a cultural centre for learning about China. |

Table 15: Examples of concrete generations for MRPC. The first sentence (*italicized*) is randomly sampled from the pretraining corpus; the second sentence (underlined) is generated by $G_{\boldsymbol{\theta}_p}$.

| Example | Label | Generated Text |
|---------|-------|----------------|
| #1 | entailment | Sentence 1: *It embodies the aesthetic ideals of the Queen Anne style through its design and detailing and shows the type of house that attracted prosperous Napans at the end of the Victorian era.*
Sentence 2: It is a fine example of the Queen Anne style of architecture through its design and detailing and shows the type of house that attracted prosperous Napans at the end of the Victorian era. |
| | not entailment | Sentence 1: *It embodies the aesthetic ideals of the Queen Anne style through its design and detailing and shows the type of house that attracted prosperous Napans at the end of the Victorian era.*
Sentence 2: The building is a fine example in this style, with an elegant facade reminiscent to those found on many grand mansions built by wealthy merchants during America's Gilded Age. |
| #2 | entailment | Sentence 1: *Crosbie ran unsuccessfully for the leadership of the Liberal Party of Newfoundland and Labrador in 1969, losing to Smallwood, and was also a candidate in the Progressive Conservative Party of Canada's 1983 leadership election, placing third.*
Sentence 2: Crosbie was a candidate in the Progressive Conservative Party of Canada's 1983 leadership election, placing third. |
| | not entailment | Sentence 1: *Crosbie ran unsuccessfully for the leadership of the Liberal Party of Newfoundland and Labrador in 1969, losing to Smallwood, and was also a candidate in the Progressive Conservative Party of Canada's 1983 leadership election, placing third.*
Sentence 2: He lost his bid as leader after he failed twice at running against John Diefenbaker. |

