# OpenReview forum: "Meta-Weighted Language Model Tuning for Augmentation-Enhanced Few-Shot Learning"
_ICLR.cc/2023/Conference — Submitted to ICLR 2023_

### Official Review · Reviewer_TXcp · 2022-10-23

**Confidence:** 4
**Correctness:** 3
**Technical Novelty And Significance:** 2
**Empirical Novelty And Significance:** 2
**Recommendation:** 5

**Clarity, Quality, Novelty And Reproducibility:**

Clarity and Quality: The paper is written in a clear and comprehensive manner for the most part

Novelty: The method presented in this paper is mostly combining existing techniques in a novel manner. For example, they combine techniques from Meta-Weight-Net (which first proposed to learn weights as hyperparameters of the objective) and data augmentation to perform more accurate text classification.

Reproducibility: The paper offers sufficient information to reproduce the method, and also the code is available in the supplementary material.


**Strength And Weaknesses:**

Strengths

+The idea of combining an autoregressive language model for data augmentation and another classification language model that performs the task, is simple and novel to some extent.

+All used modules and components are reasonable and sound, including the pre-trained models in the framework and the considered datasets/tasks. There is also some additional analysis and ablations included.

+The empirical study shows that the proposed method FewGen outperforms other SOTA methods, with and without data augmentation on the GLUE benchmark.

+The paper has a good flow, is well-written and understandable for the most part.

Weaknesses (and questions):

-There is a lack of experiments without the data augmentation done by the autoregressive language model. To have a better understanding of how the data generation part helps, the authors should provide a baseline where the model is fine-tuned without augmentation. This will offer a fair comparison to the models trained w/o augmentation in Table 1, and also can highlight the contribution of the classification-only part of the method.

-The paper states that “general idea of meta-learning is to formulate a meta objective to enable automatic learning of hyperparameters”, which is not entirely true, since that can be considered only as a subset of the meta-learning goals. In general, meta-learning is aiming to learn good representations and accrue meta-knowledge by observing batches of few-shot tasks. Therefore, it would be good if the authors can elaborate more how their method achieves this, i.e. to explain thoroughly the learning strategy, which is only briefly mentioned to be the online optimization strategy.

-The paper uses a learnable “w” as a feedforward net with 100-dimension hidden layer, but it is not explained why this is the only considered option. For instance, other options such as self-attention layers, do not seem to be considered. Since this part is important for learning the token weights in the meta-weighted objective function, the structure of this part can be further explored, for instance by providing more ablation analysis.

-The paper is lacking more qualitative analysis and discussion. It would be nice to provide more generated samples for given k-shots and for different tasks, and to discuss how they contribute to a better performance.

-It is not entirely clear how the shots are defined in the downstream tasks. Are they predefined in the scope of the datasets? This part can be more elaborated.


**Summary Of The Paper:**

This paper is exploring the few-shot capabilities for pretrained language models, by proposing a two-step method called FewGen: first to generate novel training samples from few-shot samples to augment the original set; and second to perform classification tasks by fine-tuning a classification-based language model on these generated samples. Additionally, the generation of novel samples is guided by a meta-objective which uses a weighting mechanism to choose the most important tokens w.r.t a label and produce more discriminative text samples. The model achieves better results on the GLUE benchmark, which consists of natural language understanding tasks, compared to existing few-shot methods, both with and without data augmentation steps.

**Summary Of The Review:**

This paper is interesting and proposes to smartly combine existing techniques from data augmentation and meta-learning. However, meta-learning comes a bit of a sudden and is not clearly motivated, nor the learning strategy is thoroughly explained. The authors should try to define the link between the proposed techniques, namely, data augmentation through text generation and meta-learning the weights, to have a more consistent storyline. Also, it would also strengthen the paper if the authors consider the points and concerns raised in the “weakness” part.

---

> ### Author Response · Authors · 2022-11-20
> **Response to ​​Reviewer TXcp**
>
> Thanks for your thoughtful review and feedback! As mentioned in the general response, we have updated the paper with clarifications and analyses of our meta weight learning objective, studies of the meta weighting network architecture, and more qualitative analyses as you suggest.
> We provide more detailed discussions as follows:
> 1. **Results w/o augmentation**. Since our classifier is trained in two steps, first on the few-shot samples and then on the augmented ones, only conducting the first step will be the w/o augmentation baseline, and that is exactly the LM-BFF (Man.) baseline shown in Table 1 row 4. FewGen offers consistent and significant improvements over the w/o augmentation baseline.
> 2. **Inaccurate statement regarding meta-learning and clarifications of the meta-weight learning setup**. We agree with your critique, and we have revised and clarified our motivation and goal of learning meta weights in Section 3.2. We would like to first clarify the difference between the few-shot learning setting of MAML (Finn et al.) and ours: MAML aims to meta-train the model on various few-shot tasks such that the resulting model can learn well on new tasks given a few samples, whereas we study improving few-shot learning on each individual task (i.e., not assuming meta-training data from other tasks, but only given the few-shot samples for the target task). To this end, we propose to create augmentations based on the few-shot samples so that the classifier benefits from more training data on the target task. One way is to train a generator to model the data generation probability conditioned on each label, so that the generator can be used to create novel training data. Standard generator training places equal weights on each token’s generation loss, but not every token is equally label-discriminative for the target classification task. For example, in sentiment classification, “good/bad” is more label-discriminative than “the movie”, and the former needs to be modeled more precisely; otherwise, the generator will be more prone to generating noisy samples. Therefore, we propose to automatically learn token weights using meta-learning for the generator objective so that label-discriminative information is emphasized for generator training. As shown in Section 3.2 (Analysis of Meta Weight Learning) and Appendix A, our meta weight learning strategy updates the meta weight $w^j$ based on the similarity between the gradient of $L\_{\text{disc}}$ and the gradient of $L^j\_{\text{gen}}$ — the meta weights will be higher on those tokens where optimizing their generative objective is more beneficial for minimizing the discriminative loss.
> 3. **Meta weighting network architecture**. We have added the study of using a more complicated weighting network architecture (with one extra self-attention layer) compared to the default simple feedforward network (FFN) structure in Appendix E Table 5. The generator trained with the more complicated structure does not outperform the simple one (measured by generated text accuracy and test set perplexity). This is probably because the generator PLM’s output representations are sufficiently contextualized (after many self-attention layers) and contain the information necessary for learning the token weights, thus a simple FFN as the weighting network will be enough. Using more complicated networks, on the other hand, introduces more randomly initialized new parameters which may not be learned well using the limited amount of few-shot training data. This may also be analogous to why PLM fine-tuning usually only uses a simple FFN as the task-specific head on top of the PLM representations instead of more advanced structures.
> 4. **Qualitative analyses**. Please refer to Appendix H for concrete examples of generated training data for each task. For SST-2, we present the given few-shot training samples (16 per label) in Table 8, and some generated samples in Table 9. The synthetic samples are accurate and quite different from the given training samples to serve as effective augmentations.
> 5. **Definition of few-shots**. The original GLUE tasks contain abundant training data that can be used for fully-supervised training. To simulate few-shot learning scenarios, Gao et al. created few-shot training/validation set splits containing 16 samples per label, and as mentioned in Section 4 (Downstream Tasks and Metrics), we use exactly the same few-shot training/validation set splits defined in Gao et al. so that our comparisons with baselines are fair.
>
> Thank you again for your review! Please let us know if you have any further comments.
>
> References:
> Finn et al. “Model-Agnostic Meta-Learning for Fast Adaptation of Deep Networks.” ICML 2017
> Gao et al. “Making Pre-trained Language Models Better Few-shot Learners.” ACL 2021
> Shu et al. “Meta-Weight-Net: Learning an Explicit Mapping For Sample Weighting.” NeurIPS 2019

---

> ### Author Response · Authors · 2022-12-07
> **Looking Forward to Your Reply!**
>
> Dear Reviewer TXcp,
>
> Thank you again for your valuable feedback and comments! As the discussion period is ending soon, we would greatly appreciate it if you could let us know whether you are satisfied with our response. We will be happy to address any remaining concerns.
>
> Sincerely,
> Paper3797 Authors

---

### Official Review · Reviewer_gbEj · 2022-10-23

**Confidence:** 3
**Correctness:** 4
**Technical Novelty And Significance:** 2
**Empirical Novelty And Significance:** 2
**Recommendation:** 6

**Clarity, Quality, Novelty And Reproducibility:**

The paper is clearly written and coherent to read. The reweighting objective which is meta-learned is not novel, however the authors use it for generation task which is novel.

**Strength And Weaknesses:**

Strengths:

	- The paper shows strong improvements over other baseline few-shot fine-tuning techniques (with / without use of data-augmentation).  The comparisons with data-augmentation techniques is more fair, as extra data is used during downstream fine-tuning.

	- The main technical novelty is in formulating the fine-tuning process of the generator with the meta-objective. Conditioning the generator on a weighted version of the input is a good way to generate label-coherent examples and the authors make the framework to work which is a strength.

Weakness:

	- Could the authors add the ablation where D_train and D_gen is together used for fine-tuning?  I could not find it in the paper and it would be beneficial to add this to truly understand the effectiveness of the proposed fine-tuning strategy (with temporal ensembles) for the classifier tuning stage. In Table. (2), I do find -temporal ensemble, however I am assuming it is a two step fine-tuning - first on D_train and then on D_gen without the regularizer.

	- The compared baselines are not comprehensive. The authors should add more baselines where data-augmentation is used for few-shot learning. Some of the baseline techniques from https://aclanthology.org/2021.findings-acl.84.pdf can be investigated to strengthen the contribution of the paper. While a comprehensive analysis of all the augmentations is out of scope for the paper, a few strongly performing augmentation techniques can be added as baselines to Table 1.

I would like to see some quantitative analysis on what proportion of label noise is mitigated with the use of meta-objective in training the generator? Downstream performance is definitely one indirect way to measure it, but having a quantitative analysis would strengthen the paper.

**Summary Of The Paper:**

The paper proposes a new augmentation strategy to augment few-shot samples during fine-tuning. The core idea of the paper is to use a pre-trained language model (PLM) to generate texts which can be leveraged towards fine-tuning.  The authors propose two methods in the paper to achieve this: (i) a method to fine-tune the PLM to generate appropriate texts; (ii) a technique to fine-tune the classifier using the generated texts.

**Summary Of The Review:**

Overall, the paper is well-written, has satisfactory novelty and strong results. However, I am leaning towards weak reject as I feel that the authors did not compare with enough baselines where augmentation is used. Given the strong results, I will be willing to increase my scores if the authors can : (i) provide some motivation on why these 3 augmentation techniques were chosen to compare to or add some more baselines to Table 1.  (ii) Some quantitative measurement of the mitigation of label noise on the meta-objective.

---

> ### Author Response · Authors · 2022-11-20
> **Response to ​​Reviewer gbEj (2/2)**
>
> References:
> Anaby-Tavor et al. "Do not have enough data? Deep learning to the rescue!." AAAI 2020
> Andreas, Jacob. “Good-Enough Compositional Data Augmentation.” ACL 2020
> Chen et al. “MixText: Linguistically-Informed Interpolation of Hidden Space for Semi-Supervised Text Classification.” ACL 2020
> Ding et al. “DAGA: Data Augmentation with a Generation Approach for Low-resource Tagging Tasks.” EMNLP 2020
> Feng et al. “Keep Calm and Switch On! Preserving Sentiment and Fluency in Semantic Text Exchange.” EMNLP 2019
> Feng et al. “GenAug: Data Augmentation for Finetuning Text Generators.” 2020
> Feng et al. “A Survey of Data Augmentation Approaches for NLP.” ACL 2021
> Gao et al. “Soft Contextual Data Augmentation for Neural Machine Translation.” ACL 2019
> Guo et al. “Sequence-level Mixed Sample Data Augmentation.” EMNLP 2020
> Jindal et al. “Augmenting NLP models using Latent Feature Interpolations.” COLING 2020
> Kumar et al. “Submodular Optimization-based Diverse Paraphrasing and its Effectiveness in Data Augmentation.” NAACL 2019
> Kumar et al. “Data Augmentation using Pre-trained Transformer Models.” 2020
> Nguyen et al. “Data Diversification: A Simple Strategy For Neural Machine Translation.” NeurIPS 2020
> Schick et al. “Exploiting Cloze-Questions for Few-Shot Text Classification and Natural Language Inference.” EACL 2021
> Sennrich et al. “Improving Neural Machine Translation Models with Monolingual Data.” ACL 2015
> Thakur et al. “Augmented SBERT: Data Augmentation Method for Improving Bi-Encoders for Pairwise Sentence Scoring Tasks.” NAACL 2021
> Wang et al. “SwitchOut: an Efficient Data Augmentation Algorithm for Neural Machine Translation.” EMNLP 2018
> Wei et al. “EDA: Easy Data Augmentation Techniques for Boosting Performance on Text Classification Tasks.” EMNLP 2019
> Xie et al. “Unsupervised Data Augmentation for Consistency Training.” NeurIPS 2020
> Yoo et al. “GPT3Mix: Leveraging Large-scale Language Models for Text Augmentation.” EMNLP 2021
> Zhang et al. “SeqMix: Augmenting Active Sequence Labeling via Sequence Mixup.” EMNLP 2020

---

> ### Author Response · Authors · 2022-11-20
> **Response to ​​Reviewer gbEj (1/2)**
>
> Thanks for your thoughtful review and feedback! As mentioned in the general response, we have updated the paper with a new baseline, a new ablation, and more quantitative analyses of our meta weight learning objective as you suggest.
> We provide more detailed discussions as follows:
> 1. **Ablation that fine-tunes the classifier on the combination of $D_{\text{gen}}$ and $D_{\text{train}}$**. We have added this ablation to Table 2. For this ablation, we upsample $D_{\text{train}}$ by $100\times$ so that its size is comparable with $D_{\text{gen}}$; without upsampling, the result is much worse. One potential limitation of this ablation is that the classifier receives the clean training samples from $D_{\text{train}}$ and noisy training samples from $D_{\text{gen}}$ at the same time, and the conflicting training signals due to label noise may make model training more unstable. On the contrary, our proposed two-stage classifier training method first trains the classifier exclusively on the clean set $D_{\text{train}}$, so that the classifier’s predictions are reasonably reliable and can be used to regularize model training via temporal ensembling when trained on $D_{\text{gen}}$ in the second step.
> 2. **Additional baselines**. We have added an additional baseline MixText (Chen et al.) to Table 1. We adapt the mixup operation in MixText to fit prompt-based fine-tuning by interpolating the label words instead of categorical labels, which leads to significantly improved performance on NLU tasks (e.g., Schick et al. reported that the original MixText has <35 accuracy on MNLI while our reimplemented MixText achieves >65). We provide more justifications regarding the choice of our baseline methods:
>
>     * MixText represents the family of mixup-based augmentation methods (e.g., Chen et al., Guo et al., Jindal et al., Zhang et al.). We include MixText in the comparisons because it is designed for sequence classification tasks while others are for different task types.
>     * Back translation represents paraphrase-based methods (e.g., Kumar et al. 2019, Sennrich et al., Wei et al., Xie et al.). We include back translation in the comparisons because it uses trained translation models to generate paraphrases and generally produces better results than other heuristic-based methods (e.g., synonym replacement).
>     * Generator fine-tuning is the central idea in Anaby-Tavor et al. & Kumar et al. 2020.
>     * GPT3Mix (Yoo et al.) is a strong method that uses one of the largest existing PLMs (175B GPT3) for augmentations via few-shot demonstrations.
>
>     For other methods mentioned in the survey paper (Feng et al. 2021), some apply simple alterations to training samples via word/phrase replacements, swapping and deletion (e.g., Andreas, Feng et al. 2019, Feng et al. 2020, Wei et al.), and these methods generally underperform the above baselines we have compared (as shown in Yoo et al.). There are also augmentation methods designed for specific applications such as machine translation (Gao et al., Nguyen et al., Wang et al.), sentence similarity (Thakur et al.) and token tagging (Ding et al.). These methods are not directly applicable to our studied NLU tasks. Please let us know if we still miss any important baselines, and we will be happy to discuss/compare with them.
> 3. **Quantitative measurement of label noise mitigation using the meta-objective**. We have added the quantitative analyses in Appendix G, Table 6. Using our meta weighted objective indeed results in less noisy generated samples compared to the standard generation objective, due to better modeling of label-discriminative tokens.
>
> Thank you again for your review! Please let us know if you have any further comments.

---

> > ### Comment · Reviewer_gbEj · 2022-11-21
> > **Thank you for the response**
> >
> > I thank the authors for their response. The authors have addressed the questions and added the new experiments which have made the paper better. I have updated my score.

---

> > > ### Author Response · Authors · 2022-12-07
> > > **Thank You!**
> > >
> > > Dear Reviewer gbEj,
> > >
> > > Thanks for your feedback to our paper revision and response!
> > >
> > > Sincerely,
> > > Paper3797 Authors

---

### Official Review · Reviewer_1UgY · 2022-10-24

**Confidence:** 4
**Correctness:** 2
**Technical Novelty And Significance:** 2
**Empirical Novelty And Significance:** 2
**Recommendation:** 3

**Clarity, Quality, Novelty And Reproducibility:**

Clarity, Quality, and Novelty: The paper is well-written and easy to understand. The intuition that formulates label-discriminative text generation into a meta-learning framework is not clear and a bit sudden.

Reproducibility: Code is provided in supplementary materials.




**Strength And Weaknesses:**

Strength:
1. The overall framework to investigate the few-shot learning ability of PLM  is somehow novel.
2. It is interesting to design the $L_{disc}$ to encourage the PLM-generator to synthesize more label-discriminative augmentation examples.
3. The paper is well-written with clear logic. Most part of the paper is easy to understand.


Weakness:
1. The intuition that formulates label-discriminative text generation into a bi-level problem is not clear. What is the interpretation behind it? It seems that the bi-level structure is not necessary in this case, since there is no strict hierarchy of the inner and outer objectives ($L_\text{gen}$ and $L_\text{disc}$): the order of inner and outer objectives can be simply swapped to achieve the same goal(i.e. using unweighted generation loss in the outer loop and weighted discriminative loss in the inner loop). The bi-level form here is more like to introduce “technical contribution”.
2. The two reason for not directly combining $L_\text{disc}$ and $L_\text{gen}$ is not well justified ( "the generation-irrelevant loss  $L_\text{disc}$  will unavoidably interfere with the language modeling process” and “A hyperparameter needs to be introduced to balance the weights of the two losses ”.) I do not see why the bi-level formulation helps solve these issues. Specifically, suppose the token weights are learned to minimize the discriminative loss, how can it guarantee such weights will not affect the language modeling process? In fact, even without direct intervention, the $L_\text{disc}$ will affect $w$ and therefore affect the optimization of $L_\text{gen}$. Besides, it seems that introducing a hyperparameter to balance  $L_\text{disc}$  and $L_\text{gen}$ is acceptable, as the author introduces such a hyperparameter in Eqn. 5 to balance the two loss term for finetuning, which is actually in a similar situation.
3. I remain skeptical about the performance gain of the proposed method. Firstly, the authors do not compare with the important baseline that directly uses few shots in the demonstration(as in-context examples) to generate augmentation data, and then use the labeled few shots and synthetic data to fine-tune a PLM classifier. I find that the cited paper (Meng et al., 2022) uses the same PLM generator and classifier, as well as a much restricted zero-shot setting, which can achieve comparable performance with FewGen in some task. It looks like the performance gain (if any) is quite marginal, given that few-shot should already introduce some performance boost compared with zero-shot. Besides, I am curious about to which extent more label-discriminative data can improve the moderate-size PLM classifier. Table2 should include all the glue tasks. Complete results of the important baseline(w. $L_\text{gen}$) can help justify which margin of improvement can the proposed method achieve.
4. The paper does not conduct quantitative and qualitative analysis to measure the quality of the generated text. Can the improved generator really generate more label-discriminative text? Some automatic metrics/human evaluation should be introduced to measure the quality of the generated data.
5. Usually, the bi-level formulations used in previous work present justifications of why such formulation is adopted, such as meta weight net, MAML, etc. However, no theoretical justification or intuitive explanation is given to explain the necessity and superiority of such a formulation in this paper.
6. The performance bottleneck of the proposed method is quite obvious, since it is not applicable to large PLMs, and moderate PLMs(e.g. 1.6B Crtl) usually do not have good generation ability. From the experimental details, it is quite concerning that for the experiments using Ctrl as PLM-generator, the batch size can be only set to 2 on A100 GPUs. This brings the usability of this method on larger PLM into question. Besides, it is well known that meta-learning methods are notoriously hard to tune and can be extremely sensitive to hyperparameter choices. It is hard to tune the framework on a large PLM with a small batch size.  In addition, the updating process of the generator looks time-consuming, as the gradient calculation for prefix tuning needs to backward pass all the gradients through the PLM’s parameters in each iteration.
7. The authors only conduct experiments on 2-class or 3-class NLU classification tasks. It is questionable whether the proposed method is suitable for other tasks(e.g., multi-class classification and QA tasks).


**Summary Of The Paper:**

The paper proposes a method to augment label-discriminative data in PLM few-shot learning setting. The main idea is to improve the PLM-generator by introducing a label-discriminative loss in a meta-learning framework. The improved PLM-generator is then used to generate augmentation data to finetune a classification PLM(Roberta-large). The paper compares the proposed methods with several few-shot baselines on seven classification tasks of GLUE.


**Summary Of The Review:**

Overall, the paper is well-written with a clear flow. However, (1) the intuition that formulates label-discriminative text generation into a bi-level problem is not clear, as there is no strict hierarchy of the two objectives. No convincing intuitive explanation or theoretical justification is given to explain the necessity and superiority of such a meta-framework in this paper. (2) Some important baselines are missing, and I remain skeptical about the quality improvement of generated data and performance gain of the proposed method.

---

> ### Author Response · Authors · 2022-11-20
> **Response to ​​Reviewer 1UgY (2/2)**
>
> Continued response:
>
> 5. **Bi-level optimization justification**. Please refer to Section 3.2 and our answers to Q1 & Q2 above. Our bi-level meta-weight learning framework addresses the limitation of $L\_{\text{gen}}$ which does not model the distinction across different labels (Figure 2), and also avoids directly using $L\_{\text{disc}}$ to update the generator parameters which may hurt the generation quality (Tables 6 & 7).
> 6. **Applicability to larger models**. Since we use prefix-tuning for generator training, only $<0.1\\%$ of the model parameters are updated and our method is quite scalable to large models. We use CTRL in our experiments because we aim to ensure our results are reproducible on a single GPU. Even with CTRL, our method achieves better results than GPT3Mix which uses a $100\times$ large model. We use batch size = 2 simply because NLU few-shot learning usually benefits from small batch sizes (e.g., Appendix C.1 in Gao et al. also used small batch sizes). We can set batch size up to 32 on a single GPU (i.e., train on the entire few-shot set for each batch), but that results in worse perplexity. To use larger models, one may need advanced hardware, but this may be more of a general issue of using large PLMs than a limitation of our method. For hyperparameter setup, we do not perform per-task tuning but use a fixed set of hyperparameters, which works quite well for all tasks. Our meta weight learning process (takes ~10 mins per task) indeed has higher time cost than standard training (takes ~3.5 mins per task), but the extra cost is quite affordable for few-shot learning. Please refer to Appendix C (Details of Generator Tuning).
> 7. **Other classification tasks**. Our method is generalizable to any multi-class sequence classification tasks. We evaluate on GLUE because it is a challenging benchmark whose task difficulty and coverage have been well studied and recognized, serving as good metrics for few-shot learning. For other classification tasks with a larger label space (e.g., topic classification), their difficulty is not necessarily higher than GLUE and may not benefit much from data augmentation. For example, Mukherjee et al. show that few-shot learning methods without augmentation can already achieve $>98.5\\%$ accuracy on DBPedia (14-way classification). Although our method can also be applied to these datasets, the performance gain brought by augmentation is unlikely to be notable. For sequence-level QA classification tasks (question-answer entailment), QNLI has been included in the GLUE benchmark. For token-level QA tasks (extracting answer spans) like SQuAD, our method is not applicable because they do not fit into our “given a class label, generate a text sequence” framework. These token-level QA tasks are indeed less studied in the few-shot learning literature — they also cannot be solved by the prominent prompt-based learning methods (e.g., Gao et al. and Zhang et al.). We’d like to leave those tasks as future work.
>
> Thank you again for your review! Please let us know if you have any further comments.
>
> References:
> Gao et al. “Making Pre-trained Language Models Better Few-shot Learners.” ACL 2021
> Mukherjee et al. “Uncertainty-aware Self-training for Few-shot Text Classification.” NeurIPS 2020
> Zhang et al. “Differentiable Prompt Makes Pre-trained Language Models Better Few-shot Learners.” ICLR 2022

---

> ### Author Response · Authors · 2022-11-20
> **Response to ​​Reviewer 1UgY (1/2)**
>
> Thanks for your thoughtful review and feedback! As mentioned in the general response, we have updated the paper by adding more clarifications/analyses of the meta weight learning strategy, extending the ablation results and adding more quantitative and qualitative analyses of generated texts as you suggest.
> We provide more detailed discussions as follows:
> 1. **Bi-level optimization formulation**. As shown in Eq. (4), the inner objective directly optimizes the generator’s parameters $\boldsymbol{\theta}\_p$ while the outer objective optimizes the meta weights used by the inner objective but does not directly update $\boldsymbol{\theta}\_p$. Since only $\boldsymbol{\theta}\_p$ (but not the meta weights) will be used for synthetic data generation after training, the inner objective is the “task objective” and the outer objective is the “auxiliary objective” that learns the hyperparameters for the inner objective. If the inner objective is the discriminative loss and the outer objective is the generation loss, $\boldsymbol{\theta}\_p$ will be directly optimized by $L\_{\text{disc}}$ but not $L\_{\text{gen}}$, and will not be suitable for text generation.
> 2. **Why not combine $L\_{\text{disc}}$ and $L\_{\text{gen}}$**? Using $L\_{\text{disc}}+L\_{\text{gen}}$ is a multi-task formulation where the generator is trained to perform both the discriminative and the generative tasks. However, since the generator will eventually be used only for generation, the discriminative loss interferes with the generation task, resulting in worse language modeling quality. Eq. (5) is a different case because both terms in Eq. (5) optimize the same classification probability, thus it is not multi-task learning but learning with regularization. Our proposed $L\_{\text{w-gen}}$ also changes the standard language modeling process, but in a different way: Instead of directly using $L\_{\text{disc}}$ to update the generator parameter, $L\_{\text{w-gen}}$ uses $L\_{\text{disc}}$ to reweight the contributions from different tokens to the generation loss. Such a design is motivated by the observation that not every token is equally label-discriminative in NLU tasks. For example, in sentiment classification, “good/bad” is more label-discriminative than “the movie”, and the former needs to be modeled more precisely; otherwise, the generator will be more prone to generating noisy samples. Also given that the generator has been pretrained and is already good at modeling “easy” tokens (e.g., stopwords, punctuation), equally weighting each token for generator tuning may not be optimal, and it could be beneficial to focus on more important tokens for the classification task by assigning them higher weights. As shown in Section 3.2 (Analysis of Meta Weight Learning), our method achieves this by updating the meta weight $w^j$ based on the similarity between the gradient of $L\_{\text{disc}}$ and the gradient of $L^j\_{\text{gen}}$ — the meta weights will be higher on those tokens where optimizing their generative objective is more beneficial for minimizing the discriminative loss. This allows the generator to effectively capture the nuanced differences across labels and leads to better language modeling quality (as shown in Table 6, $L\_{\text{w-gen}}$ induces the lowest perplexity on the test set).
> 3. **Baseline & ablation comparisons**. The GPT3Mix baseline is exactly a few-shot demonstration method for augmentation, and it uses a much larger generator (175B GPT3). Please refer to Section 5.1 (Comparison with GPT3Mix) for detailed discussions. In short, few-shot demonstration can only use a small subset of the few-shot samples at a time for creating each augmentation, as the number of demonstrations received by the model is bounded by its maximum input sequence length. Regarding Meng et al., they used COCO-LM as the classification PLM while we used RoBERTa to ensure our comparisons with baselines are fair. Table 5 in Meng et al. showed that replacing COCO-LM with RoBERTa generally worsened the results. Also, Meng et al. required generating much more training data than needed (e.g., $10\times$) and only used the top-ranked ones based on generation probability for training, incurring much longer data generation time than our method. Regarding ablation results, we have extended them to all tasks (Table 2).
> 4. **Quantitative & qualitative analysis**. Please refer to Appendices G and H for the added analyses. Our proposed meta-weighted generator tuning objective has better language modeling quality (measured by perplexity) and generates accurate training data that can effectively augment the few-shot training set.

---

> ### Author Response · Authors · 2022-12-07
> **Looking Forward to Your Reply!**
>
> Dear Reviewer 1UgY,
>
> Thank you again for your valuable feedback and comments! As the discussion period is ending soon, we would greatly appreciate it if you could let us know whether you are satisfied with our response. We will be happy to address any remaining concerns.
>
> Sincerely,
> Paper3797 Authors

---

### Official Review · Reviewer_KP32 · 2022-10-24

**Confidence:** 3
**Correctness:** 3
**Technical Novelty And Significance:** 2
**Empirical Novelty And Significance:** 2
**Recommendation:** 6

**Clarity, Quality, Novelty And Reproducibility:**

**Clarity**
The narrative of the manuscript is clear and pleasure to read - the problem is clearly described and every piece of the overall proposed method is well motivated and explained.

**Quality**
The proposed method looks sound to me and it intuitively makes sense. Experimental results and ablation studies confirm the validity of the proposed method. As mentioned above, adding more qualitative evaluations will make the contribution stronger.

**Novelty**
Although individual components might not be sufficiently novel, the overall contribution (i.e. the end-to-end algorithm) is novel and it cleverly combines and modifies existing method to solve the desired problem. I have no concerns regarding the novelty.

**Reproducibility**
Authors provide sufficient details and pseudo-code corresponding to their methods. As mentioned above, some additional details on the stability and ease-of-training of the meta-learning procedure will be helpful (apologies if I have missed it).

**Strength And Weaknesses:**

**Strengths**
* The paper is enjoyable to read and the algorithm has been developed nicely throughout the narratives. The authors clearly describe the problem they are intending to solve followed by motivating and describing the three major components of their algorithm - the discriminative loss, meta-weighted generative loss and the ensembled classifier. Authors also provide the end-to-end algorithms which help the reader to understand and possibly implement the proposed method.
* Experimental results on the GLUE benchmark show clear and non-trivial improvement over existing baselines including the ones that rely on generating additional samples and those which do not.
* Although the final proposed method is somewhat complicated, ablation studies on the importance of each of the component show relative improvement caused by individual components and the importance of all these pieces.
Overall, I find it to be a strong paper with a non-trivial contribution, clear narrative and strong empirical results.

**Weaknesses/Comments**
* Qualitative analysis - Given that it is a generative paper, I'd like to see examples of how the generated texts look like for different labels. How different are those from the ones present in the training sample? On the same line, I'd like to see how well the meta-weight generation can learn the importance of various tokens. For example, authors provide an example that the token "movie" might not be relevant to generate texts for positive/negative movie reviews. Does such properties automatically emerge from the meta-learning procedure?
* Susceptibility to hyper-parameter tuning - The optimization procedure is performing three interleaved gradient updates at every iteration. I'd imagine that such an optimization procedure will require sophisticated hyper-parameter tuning to obtain stable convergence. I'd like authors to comment on that along with the overall computational cost for this procedure.
* The whole generator-discriminator paradigm is very similar to many works from the literature which are based on GANs. I'd like the authors to clarify in the paper how their optimization process varies differently from the GAN.
* Generalizability of the ensembled classifier - Can the ensembled classifier generalize to other methods which also produce synthetic samples to augment training set? If so, how much gain can we tentatively expect to obtain with those methods?


**Summary Of The Paper:**

In this work, authors propose a text-generation method to augment training data for various discriminative few-shot NLP tasks (from the Glue benchmark) such that the model performs better on these tasks as opposed to training (e.g. prompt-tuning/fine-tuning) using the original train split alone. Towards this end, authors propose i) a discriminative loss to ensure generated texts produce enough diversity to differentiate between labels and ii) a meta-weighted generative loss to take into account the impact of different tokens towards a label. They also propose an ensembling classifier to add more robustness to deal with generated samples. Experimental results on various tasks from the GLUE benchmark shows that the proposed method outperforms other methods from the literature.

**Summary Of The Review:**

Overall, I found the paper to have a solid technical contribution to solve the task at hand which is to improve few-shot performance of various discriminative NLP tasks by learning to generate diverse synthetic samples. Experimental results and ablation studies showcase the efficacy of the algorithm. At this point, I am inclined to accept this paper based on my opinion alone.

---

> ### Author Response · Authors · 2022-11-20
> **Response to ​​Reviewer KP32**
>
> Thanks for your thoughtful review and feedback! As mentioned in the general response, we have updated the paper with more qualitative analyses and the time cost/hyperparameter setup for generator tuning as you suggest.
> We provide more detailed discussions as follows:
> 1. **Qualitative analyses**. Please refer to Appendix H for concrete examples of generated training data for each task. For SST-2, we present the few-shot training samples (16 per label) in Table 8, and some generated samples in Table 9. The synthetic samples are accurate and quite different from the given training samples to serve as effective augmentations. For other sequence-pair tasks, because we randomly sample the first sequence from the pretraining corpus (e.g., Wikipedia) and let the generator create the second sequence given certain labels, the resulting generated samples will be certainly different from the given training samples. For case studies regarding learned token weights, please refer to Appendix E Figure 5. The tokens with higher meta weights are generally more label-discriminative.
> 2. **Hyperparameter tuning and time cost**. Please refer to Appendix C, Details of Generator Tuning. We do not perform task-specific hyperparameter tuning but use a fixed set of hyperparameters, which works quite well for all tasks in our experiments (As shown in Table 5, our proposed meta-weighted generator tuning method has consistently lower perplexity on the test set than standard generator tuning). We also do not observe convergence/instability issues, probably because our meta weighting network is a lightweight 1-layer feedforward network which is easy to optimize. Our meta-weighted generator tuning (takes ~10 mins per task) indeed has a higher time cost than standard tuning (takes ~3.5 mins per task), but the extra time cost is affordable due to the small few-shot training set.
> 3. **Differences from GANs**. The major difference is that GANs jointly train the generator and discriminator networks in an adversarial manner (the generator tries to fool the discriminator), whereas in our framework, the generator PLM is trained independently from the classification PLM. Such differences are related to the difficulty of applying GANs for training text generators. Previous studies (Caccia et al.) show that GAN-style text generator training may not produce better quality and diversity than standard maximum likelihood training. Therefore, for simplicity and effectiveness, we do not use GAN-style generator training, but it will be an interesting future work direction to explore joint training of generator and classification models for text generation/augmentation.
> 4. **Generalizability of temporal ensembling for classifier training**. Since temporal ensembling is a regularization technique that improves the noise-robustness of classifier training and is independent of how synthetic samples are created, we believe it can be also applied to other augmentation methods. For example, the first two ablations presented in Table 2 also use temporal ensembling for classifier training, and their results will be worse without it (the gap is similar to that between FewGen and -temporal ensemble).
>
> Thank you again for your review! Please let us know if you have any further comments.
>
> Reference:
> Caccia et al. “Language GANs Falling Short.” ICLR 2020

---

> ### Author Response · Authors · 2022-12-07
> **Looking Forward to Your Reply!**
>
> Dear Reviewer KP32,
>
> Thank you again for your valuable feedback and comments! As the discussion period is ending soon, we would greatly appreciate it if you could let us know whether you are satisfied with our response. We will be happy to address any remaining concerns.
>
> Sincerely,
> Paper3797 Authors

---

> > ### Comment · Reviewer_KP32 · 2022-12-07
> > **Apologies for delay in response**
> >
> > Hi, I apologize for delay in replying. I am happy with the responses. I also checked a few generated samples and they make sense and in line with my expectations.

---

### Author Response · Authors · 2022-11-20
**General Response**

We sincerely thank the four reviewers for thoughtful and detailed comments! We have updated the paper according to reviewers' suggestions, and we summarize the major changes as follows (also highlighted with blue color in the updated version):
1. Clarify the motivation and formulation of meta weight learning (Section 3.2) per Reviewers 1UgY and TXcp.
2. Analyze the meta weight learning strategy (Section 3.2, Analysis of Meta Weight Learning & Appendix A) per Reviewers 1UgY and TXcp.
3. Add another augmentation baseline MixText (Table 1) per Reviewer gbEj.
4. Extend the ablation study results to all tasks (Table 2) per Reviewer 1UgY.
5. Add another ablation that fine-tunes the classifier on the combination of $D_{\text{gen}}$ and $D_{\text{train}}$ (Table 2) per Reviewer gbEj.
6. Add details about the time cost and hyperparameter setup for generator tuning (Appendix C) per Reviewers KP32 and 1UgY.
7. Study different meta weighting network architectures (Appendix E) per Reviewer TXcp.
8. Add quantitative analyses of generated text accuracy and generator perplexity (Appendix G) per Reviewers 1UgY and gbEj.
9. Add qualitative analyses for concrete generation results (Appendix H) per Reviewers KP32, 1UgY and TXcp.

Sincerely,
Paper3797 Authors

---

### Decision · Program_Chairs · 2023-01-20

**Decision:**

Reject

**Justification For Why Not Higher Score:**

See metareview for concerns.

**Justification For Why Not Lower Score:**

N/A

**Metareview: Summary, Strengths And Weaknesses:**

This paper proposes a technique for improving few-shot fine-tuning performance by using the few-shot examples to induce a separate language model to generate additional training samples. The auxiliary autoregressive LM is fine-tuned on the few-shot examples using prefix tuning and a novel loss and optimization procedure that encourages producing text that is consistent with a label. While the proposed method demonstrated improved performance over prior work, reviewers had various concerns, including the complexity of the method (which is especially concerning in few-shot settings where limited data is available for system tuning), the motivation of bi-level optimization, and insufficient baselines. The consensus was therefore to reject this paper.